# Revealing The Intrinsic Ability of Generative Text Summarizers for Outlier Paragraph Detection

## Abstract

Generative text summarizers are good at content encapsulation but falter when outlier paragraphs disrupt the primary narrative. We categorize these outliers into **cross-document** outliers that are thematically inconsistent but within the same domain, and **cross-domain** outliers, originating from distinct domains. Traditional methods lean on word embeddings and specialized classifiers, requiring extensive supervised fine-tuning. Confidence-based strategies, despite bypassing fine-tuning, are ill-suited due to the non-classification essence of summarization. Leveraging the encoder-decoder cross-attention framework, we introduce an approach emphasizing the unique characteristics of infrequent words in detection. We present CODE, a novel outlier detector using a closed-form expression rooted in cross-attention scores. Our experimental results validate the superiority of CODE under different datasets and architectures, e.g., achieving a 5.80% FPR at 95% TPR vs. 25.63% by supervised baselines on the T5-Large and Delve domain. We further underscore the significance of cross-attention, word frequency normalization and judicious integration of cross-document outliers during pretraining.[1]

## 1 Introduction

Generative text summarizers efficiently distill vast content into concise summaries (See et al., 2017; Liu & Lapata, 2019; Radford et al., 2019; Brown et al., 2020; OpenAI, 2022). While adept at capturing coherent sequences, these models struggle with outlier paragraphs interspersed within the content. This limitation can distort the integrity and quality of the generated summary (Liu et al., 2020; Tan et al., 2017). For instance, blending a paragraph on aquatic ecosystems into a desert-focused sequence can produce a misleading and incongruous summary.

In this paper, we address the outlier paragraph detection, distinguishing between two main types, i.e., **cross-document** and **cross-domain**. Cross-document outliers deviate thematically, even within the same domain, while cross-domain outliers come from entirely different domains. For example, a paragraph on aquatic plant evolution in an article on marine animals is a cross-document outlier, while a section on quantum physics in the same piece is a cross-domain outlier.

A common approach is using the supervised methods, i.e., extracting embeddings from the summarizer and fusing them with a detection classifier (Lewis et al., 2019; Li et al., 2022). This requires extensive fine-tuning after the initial pre-training (Devlin et al., 2018; Yang et al., 2019), especially with large language models. Confidence-based methods, suitable for out-of-distribution tasks (Hendrycks & Gimpel, 2016; Hsu et al., 2020), sidestep fine-tuning by assigning a confidence score to classification results but do not naturally fit text summarization, since it is not standard classification task. Moreover, unlike out-of-distribution tasks that evaluate entire sequences, outlier detection evaluates individual paragraphs for coherence.

In this paper, we investigate generative text summarizers built upon the encoder-decoder cross-attention architecture (Vaswani et al., 2017). Our preliminary observation suggests that infrequent words often exhibit domain-specific characteristics, which can potentially pinpoint their source domain. Notably, during pre-training with cross-document outlier paragraphs, such infrequent words

---

[1]Our code is available at: `https://anonymous.4open.science/r/code-B649/`

in outlier paragraphs typically garner reduced cross-attention scores with the generated summary. Conversely, those in coherent paragraphs tend to see elevated cross-attention scores. Leveraging these observations, we introduce a method for pre-training text summarizers that incorporate cross-document outlier paragraphs. Furthermore, we present CODE (Cross-Attention Outlier Detector), an innovative anomaly detection mechanism that employs a closed-form expression based on the cross-attention scores in generative language models. The core contributions of this paper are:

- Proposal of a method to pre-train generative text summarizers incorporating cross-document outliers. We subsequently introduce the CODE detector, which computes average cross-attention scores, normalized by word occurrences, between the generated summary and each paragraph in the sequence.

- Introduction of data pipelines to devise four pre-training datasets integrated with cross-document outlier paragraphs. Additionally, we present four cross-document outlier detection datasets and sixteen cross-domain outlier detection datasets. Our method consistently surpasses two supervised baselines across three metrics. Notably, CODE achieves a 5.80% FPR at a 95% TPR, in contrast to the 25.63% marked by the supervised fine-tuning baseline on the T5-large architecture within the Delve domain.

- An ablation study underscoring the impact of cross-attention, word frequency normalization, and the incorporation of cross-document outliers during pre-training.

The remainder of this paper is organized as follows: The problem is formulated in Section 2. Our proposed methodology is detailed in Section 3. The data pipelines and datasets are introduced in Section 4. Experimental results are presented in Section 5, followed by discussions in Section 6. We draw our conclusions in Section 7. Additional results are included in Appendix.

## 2 Preliminaries and Problem Formulation

**Text Summarizers Pretrained with Cross-document Outliers.** Let the paragraph $X$ be a word sequence contained within a document $D$, where the document is drawn from a domain set $\mathcal{D}$. Let $\mathcal{X}$ denote a paragraph sequence for summarization. For example, if $\mathcal{D}$ represents the domain containing all research papers, each paper serves as a document and $\mathcal{X}$ consists of paragraphs from their introductions. We refer to the domain $\mathcal{D}$ as the *text summarization domain*. We note that paragraphs within $\mathcal{X}$ may originate from different documents. Let the word sequence $Y(\mathcal{X})$ denote the summary of a paragraph set $\mathcal{X}$. Let $\mathbb{P}(X|D)$ denote a paragraph sampling distribution defined on the document $D$. Let $\mathcal{C} = \{(\mathcal{X}_i, Y_i)\}_{i=1}^n$ represent the pre-training set for text summarization. Each paragraph in the sequence $\mathcal{X}_i$ is drawn from an underlying mixed paragraph distribution $P(X|D_i, D_i')$, associated with two distinguished and unknown documents $D_i$ and $D_i'$,

$$P(X|D_i, D_i') = (1 - \varepsilon_i) \, P(X|D_i) + \varepsilon_i P(X|D_i'),$$

where a small and non-negative number $\varepsilon_i$ denotes the unknown probability that paragraphs in the set $\mathcal{X}_i$ are drawn from the document $D_i'$. We refer to paragraphs in $\mathcal{X}_i \cap D_i'$ as **cross-document outlier** paragraphs, and those in $\mathcal{X}_i \cap D_i$ as **coherent** paragraphs. This implies that most paragraphs in $\mathcal{X}_i$ originate from document $D_i$, creating coherence, while outlier paragraphs come from a different document $D_i'$. We use "cross-document" to indicate that both coherent and outlier paragraphs are sampled from the same domain but in different documents, distinguishing this from the outlier detection problem where outlier paragraphs may originate from different domains.

A text summarizer $G$ processes the paragraph set $\mathcal{X}$ to produce a summary $\hat{Y}(\mathcal{X})$. We employ the generative language model (**GLM**) for this task. Given $\mathcal{X}$, the summarizer $G$ generates a summary sequence $\hat{Y} = (\hat{y}_0, ..., \hat{y}_t, \hat{y}_{t+1}, ...)$ in an iterative way. The initial word $\hat{y}_0$ is a special word that signals the model to begin text generation. At each step $t$, the summarizer $G$ takes the word $\hat{y}_t$ as input to generate the next word $\hat{y}_{t+1}$, i.e., $\hat{y}_{t+1} = G(\hat{y}_0, ..., \hat{y}_t; \mathcal{X})$. We use the function $g$ denote the output of all neurons in the summarizer $G$. Specifically, we use the vector $g(\hat{y}_0, ..., \hat{y}_t; \mathcal{X})$ to denote all neuron outputs in $G$ when generating word $y_{t+1}$ at time $t$ with input $\mathcal{X}$. Let $g(\mathcal{X})$ represent the vector containing all neuron outputs in the generative model when generating sequence $\hat{Y}$. In the text summarization, we train the generative language model $G$ to ensure that the generated $\hat{Y}(\mathcal{X}_i)$ aligns with the ground truth summary $Y_i$ for all samples in the training set $\mathcal{C}$. As mentioned earlier,

each paragraph set in the set $\mathcal{C}$ contains cross-document outliers, hence we refer to the summarizer $G$ pre-trained with cross-document outliers.

**GLM-based Outlier Pargraph Detection Problem.** Let the generative model $G$ be a text summarizer pre-trained on the pre-training set $\mathcal{C}$. We are now investigating the transferability in the outlier paragraph detection problem. Let $\mathcal{D}_{\text{out}}$ denote the outlier domain from which the outlier paragraphs are drawn. Consider $\mathcal{U}$ as a paragraph sequence where coherent paragraphs in this sequence come from the text summarization domain $\mathcal{D}$, while outlier paragraphs are from the outlier domain $\mathcal{D}_{\text{out}}$. For $\mathcal{U}$, we use the binary vector $V \in \{0,1\}^{|\mathcal{U}|}$ as the label vector, where $V_i$ equals 1 if the $i$-th paragraph in $\mathcal{U}$ is an outlier paragraph and 0 otherwise. Let $\mathbb{P}_{\mathcal{U} \times V}$ denote the joint distribution of paragraph sequence $\mathcal{U}$ and label vector $V$. Let $\mathcal{C}_{\text{detect}} = \{(\mathcal{U}_k, V_k)\}_{k=1}^{m}$ be the training set for the outlier paragraph detection. Notably, we allow the outlier domain to be the same as the text summarization domain, where coherent and outlier paragraphs originate from different documents. In this scenario, it is referred to as **cross-document** outlier detection. If the outlier domain differs from the text summarization domain, it is referred to as **cross-domain** outlier detection.

A classifier $f_\theta$ utilizes neuron outputs $g$ from the text summarizer $G$ to predict the label vector. Here, the vector $\theta$ encompasses all parameters in the classifier $f_\theta$. Consequently, a GLM-based outlier paragraph detector consists of the composition of classifier $f_\theta$ and neuron output function $g$, i.e., $f_\theta \circ g$. The predictor takes the paragraph sequence $\mathcal{U}$ as input and produces an $|\mathcal{U}|$-dimensional vector, where the $k$-th element in this output vector predicts whether the $k$-th paragraph in the sequence $\mathcal{U}$ is an outlier paragraph or not. Let $\ell$ denote the cross-entropy loss function, evaluating the difference between the output of the GLM-based relation predictor $f_\theta \circ g(\mathcal{U})$ and ground truth $V$. During training, a set of parameters $\theta$ is chosen such that the following empirical loss $L_n(\theta)$ is minimized,

$$L_n(\theta) \triangleq \frac{1}{|\mathcal{C}_{\text{detect}}|} \sum_{k=1}^{m} \ell(f_\theta \circ g(\mathcal{U}_k), V_k). \tag{1}$$

During testing, the GLM-based outlier detector $f_\theta \circ g$ is evaluated on the following testing loss to determine whether it is capable of generalizing or not, $L(\theta) = \mathbb{E}_{\mathcal{U} \times v} \left[ \ell(f_\theta \circ g(\mathcal{U}), V) \right]$.

# 3 GLM-BASED OUTLIER PARAGRAPH DETECTOR

In this paper, we primarily focus on generative language models using the Transformer encoder-decoder architecture (Vaswani et al., 2017), specifically **BART** (Lewis et al., 2019) and **T5** (Raffel et al., 2020). To see the influence of the model size, we select BART-Base, BART-Large, T5-Base and T5-Large. We pre-train all GLMs on each of the pre-training sets introduced in the next section.

## 3.1 BASELINES

We concatenate the neuron outputs inside the GLM with a multi-layer perception to construct two supervised baselines. Given the potentially large number of neurons in GLMs, to reduce the computational complexity, we streamline the computation by using the input from the last encoder-decoder attention layer as the input to the multi-layer perceptron.

**Frozen.** First, we feed a paragraph sequence into the GLM and obtain a generated summary. Probing the input of the last encoder-decoder attention layer, we obtain the word embeddings of the paragraph sequence from the encoder, as well as the word embeddings of the corresponding summary from the decoder. Second, to get the embeddings of the entire sequence of the paragraph or summary, we perform a mean pooling on the obtained word embeddings that are also adopted in references (Reimers & Gurevych, 2019; Gao et al., 2021). Finally, we feed the word embedding into a multi-layer fully-connected ReLU network to detect the outlier paragraphs in the input sequence. In the supervised training phase, we freeze all parameters of the pre-trained GLM and only fine-tune the parameters of the fully connected ReLU network.

**Finetuning-all (FT-ALL).** We adopt the same architecture used in the previous baseline for outlier detection. The only difference lies in the training stage, where the parameters of the pre-trained GLM are fine-tuned along with FNN parameters.

## 3.2 CODE: Cross Attention-based Outlier paragraph DEtector

Both baselines presented in the previous section require further fine-tuning, which is usually time-consuming, especially when we need to finetune all parameters in a large GLM. In this section, we propose CODE, which eliminates the need for further fine-tuning once the GLM is pre-trained. Similar to the baseline, we also probe the input embedding of the last encoder-decoder attention layer. But, for each paragraph, we only calculate a closed-form metric based on the embedding and compare it with a threshold to determine whether the paragraph is an outlier or not.

Now we formally present our method. We concatenate all paragraphs $\mathcal{X} = \{X_1, ..., X_m\}$ and input at once to the text summarizer $G$. The GLM $G$ outputs a summary $\hat{Y}$. We input each word $\hat{y}$ in the summary $\hat{Y}$ to the decoder independently. Now we get a cross-attention matrix between the generated summary and concatenated paragraphs. When the cross attention layer has multi-head (Vaswani et al., 2017) and each head is equipped with a unique attention matrix of the same size, we average all attention matrices across different heads into one matrix. For each word $x$ in the concatenated paragraph sequence and each word $\hat{y}$ in the summary sentence $\hat{Y}$, let $Att(\hat{y}, x) \in [0, 1]$ denote the attention score in the attention matrix between the word $\hat{y}$ and $x$. We use $\frac{1}{|\hat{Y}|} \sum_{\hat{y} \in \hat{Y}} Att(\hat{y}, x)$ to measure the relevance between word $x$ and generated summary $\hat{Y}$. Let $p(x)$ denote the word frequency of $x \in X$ across all paragraphs in the training partition of the outlier detection set. We use $\frac{1}{p(x)}$ to assign more weights to the contribution of less frequent words. We define the relevance score $r(\hat{Y}, X_i) \in \mathbb{R}_+$ between the generated summary $\hat{Y}$ and the $i$-th paragraph $X_i$ as follow,

$$r(\hat{Y}, X_i) = \frac{1}{|X_i|} \sum_{x \in X_i} \frac{1}{p^\beta(x)} \left[ \frac{1}{|\hat{Y}|} \sum_{\hat{y} \in \hat{Y}} Att^\alpha(\hat{y}, x) \right] \tag{2}$$

Hyper-parameters $\alpha$ and $\beta$ are used to control the contribution of the attention score and word frequency in calculating the relevance. For a given threshold $\delta$, we say that the paragraph $X_i$ is an outlier paragraph if $r(\hat{Y}, X_i) \leq \delta$ and it is a coherent paragraph, otherwise.

## 4 Datasets

### 4.1 Data Pipeline

**Pipeline for Pre-training with Cross-document Outliers.** The source text summarization dataset includes coherent paragraph sequences and their corresponding summaries. To create a text summarization dataset with cross-document outliers, we employ a two-phase data pipeline. In the *coherent paragraph splitting* phase, we select a sample $(\mathcal{X}, Y)$ from the source dataset, where $\mathcal{X}$ represents a paragraph sequence and $Y$ is its summary. We then randomly split the sequence $\mathcal{X}$ into two sequences, denoted as $\mathcal{X} = (X_1, X_2)$. We regard these two paragraphs as coherent paragraphs. Next, in the *outlier paragraph injection* phase, we first randomly select two outlier paragraphs $Z_1$ and $Z_2$ from another two different paragraph sequences. These outlier paragraphs are randomly at three positions: before $X_1$, between $X_1$ and $X_2$ and after $X_2$. After injection, the paragraph sequence, along with the summary $Y$, constitutes a sample in our pre-training set. We note here that all outlier paragraphs in the pre-training set are cross-document paragraphs since both coherent and outlier paragraphs are sourced from the same source dataset but from different documents.

**Pipeline for Outlier Detection.** We employ the same pipeline to create outlier detection datasets. The only difference is that the outlier detection dataset does not contain the ground truth summary. In the cross-document outlier detection task, we sample the outlier paragraphs from the same source text summarization dataset, while in the cross-domain outlier detection task, we sample the outlier paragraphs from a different source dataset.

### 4.2 Pre-training Datasets with Cross-document Outliers

We choose four source datasets: CNN/Daily Mail, SAMSum, Delve and S2orc to build our pre-training dataset. The first dataset comes from the news domain, the second from dialogues, and the last two belong to the academic domain. When selecting cross-document outlier paragraphs, we

ensure that coherent and outlier paragraphs are from the same domain. Next, we present the details of each pre-training dataset.

**CNN/Daily Mail-PT dataset** is transformed from the original summarization dataset, CNN/Daily Mail (Nallapati et al., 2016) and comprises news articles sourced from CNN and Daily Mail websites, along with their human-annotated summaries. **SAMSum-PT dataset** is derived from the original text summarization dataset SAMSum (Gliwa et al., 2019) and contains dialogues with summaries constructed from existing datasets and linguists. **Delve-PT dataset** is transformed from the original summarization dataset Delve dataset (Akujuobi & Zhang, 2017; Chen et al., 2021) and consists of abstract paragraphs along with their corresponding summaries within the field of computer science. **S2orc-PT dataset** is transformed from the original summarization dataset S2orc dataset (Lo et al., 2019; Chen et al., 2021) and contains the abstract paragraphs and corresponding summaries in nineteen fields, including biology, physics, mathematics, etc.

Each data sample in the above pre-training datasets contains two coherent paragraphs, two outlier paragraphs, and one summary. The dataset partitioning is shown in Table 1. See Appendix B for the detailed statistics and construction method of each pre-training dataset.

### 4.3 Outlier Paragraph Detection Datasets

We provide an overview of the cross-document and cross-domain outlier paragraph detection datasets in the following.

**Cross-document outlier detection sets** consist of coherent and outlier paragraphs sampled from the same domain. We get four cross-document detection datasets from CNN/Daily Mail, SAMSum, Delve and S2orc, respectively.

**Cross-domain outlier detection sets** comprise coherent and outlier paragraphs from varying domains. For each domain

Table 1: The major statistics of datasets. ∗ indicates shared validation set or test set.

| Dataset | Training | Validation | Test |
|---------|----------|------------|------|
| CNN/Daily Mail-PT | 42.387K | 5.298K | 5.298K |
| SAMSum-PT | 3.273K | 0.409K | 0.409K |
| Delve-PT | 8K | 1K | 1K |
| S2orc-PT | 20K | 2K | 2K |
| CNN/Daily Mail-OD | 20K | 2.5K | 2.5K×5 |
| SAMSum-OD | 3.273K | 0.409K | 0.409K×5 |
| Delve-OD (1K) | 1K | 100* | 1K×5* |
| Delve-OD (8K) | 8K | | |
| S2orc-OD | 2K | 200 | 2K×5 |

from which coherent paragraphs are sourced, outliers are extracted from the other three domains, leading to three unique cross-domain test sets. To assess detection against random outliers, we create a set with randomly generated paragraphs using words tokenized from four summarization datasets. This results in four cross-domain test sets for each text summarization domain. Each cross-domain test set size is consistent with the cross-document set, and both types share the same training and validation datasets. In cross-domain detection, hyper-parameter tuning is exclusively done on cross-document outliers, precluding prior knowledge of cross-domain outliers during testing.

Each data sample in the above outlier paragraph detection datasets (OD) contains two coherent paragraphs and two outlier paragraphs. The dataset partitioning is presented in Table 1. Each outlier paragraph detection dataset contains a cross-document training set, a cross-document validation set, a cross-document test set and four cross-domain test sets.

## 5 Experiments

### 5.1 Experimental Setups

**Pre-training Summerizers.** We employ Hugging Face Transformers[2] Wolf et al. (2020) and AdamW optimizer with default parameters. Additional pre-training details are in the Appendix C.1. We select the checkpoint with the lowest evaluation loss for outlier detection. Generative quality is assessed using ROUGE Mihalcea & Tarau (2004), with results in the Table 7 in Appendix C.2.

**Baselines.** We employ a three-layer FNN with ReLU neurons. The input dimension $N$ is twice the dimension of the attention layer. The dimension of the first, second, and third layer is $4N$, $2N$ and

---

[2]https://huggingface.co/

Table 2: Evaluation results of CODE and baselines for cross-document outlier paragraph detection. All values are percentages. ↑ indicates that larger values are better, and ↓ indicates that smaller values are better. Characters "B" and "L" denote the Base and Large models, respectively. The hyper-parameters $\alpha$ and $\beta$ of CODE are searched by minimizing FPR at 95% TPR, and detail can be found in Table 11 in Appendix D.

| | Models | FPR (95%) TPR (↓) | AUROC (↑) | AUPR (↑) |
|---|---|---|---|---|
| | | **CODE/Frozen/FT-All** | | |
| Delve (1K) | T5-L | **5.80**/30.30/25.63 | **98.08**/92.87/94.59 | **97.03**/93.57/92.60 |
| | T5-B | **32.30**/65.97/57.75 | **90.08**/84.52/85.21 | **83.76**/82.62/82.92 |
| Delve (8K) | T5-L | **5.55**/16.85/18.28 | **98.16**/93.62/95.87 | **97.23**/94.01/95.18 |
| | T5-B | **31.50**/60.22/47.98 | **90.36**/86.32/87.64 | **84.34**/85.40/87.49 |
| S2orc | T5-L | **1.08**/10.40/6.05 | **99.54**/96.01/97.69 | **99.27**/95.59/97.32 |
| | T5-B | **2.53**/15.82/11.65 | **99.00**/96.68/96.87 | **97.95**/96.51/96.01 |
| SAMsum | T5-L | **0.60**/5.50/0.65 | **99.87**/98.67/99.68 | **99.87**/98.78/98.60 |
| | T5-B | **0.61**/8.44/1.22 | **99.66**/99.21/97.46 | **99.43**/99.00/96.68 |
| CNN/Daily Mail | T5-L | **0.00**/0.20/0.32 | **99.99**/99.85/99.77 | **99.99**/99.81/99.79 |
| | T5-B | **0.12**/0.82/0.29 | **99.96**/99.62/99.80 | **99.96**/99.56/99.70 |

$N$, respectively. We utilize the AdamW optimizer to fine-tune the model and choose the model with the lowest validation loss for testing. Training setup details are reported in Appendix C.3.

**CODE.** There are two hyper-parameters $\alpha$ and $\beta$ in CODE. We note that our method does not employ any fine-tuning in the detection phase, except that we run the hyper-parameter tuning on $\alpha$ and $\beta$. Thus, CODE is deterministic and does not have standard deviations. We search the hyper-parameters $\alpha$ in the range $[0, 2]$ with an interval of 0.1 and $\beta$ in the range $[0, 2]$ with an interval of 0.2. This implies that we search for the best setting in 231 hyper-parameter combinations. We select the model with the lowest FPR at 95% TPR for testing.

## 5.2 MAIN RESULTS

In this subsection, we present the main results. Please refer to Appendix D for further details.

**CODE vs. Baseline.** Figure 1 displays ROC curves for CODE (blue) and the baseline Frozen (red) using the T5-Large architecture on the cross-document outlier detection dataset Delve (1K). A substantial performance gap is evident, with CODE significantly outperforming the baseline. For instance, at a 95% True Positive Rate (TPR), CODE reduces the False Positive Rate (FPR) from 30.3% to 5.8%. Comprehensive evaluation results can be found in Table 2, highlighting that CODE consistently outperforms the baselines across all settings.

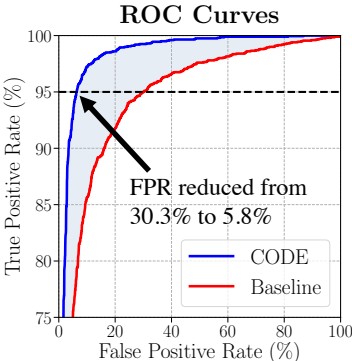

Figure 1: The ROC curves of CODE (blue) and Frozen (red) evaluated on T5-Large and Delve-OD (1K).

**Fine-tuning Dataset Size.** To assess the impact of fine-tuning dataset size, we conducted experiments on Delve using various set sizes. Interestingly, we observed that CODE exhibits low sensitivity to the set size, with consistent performance, such as a 5.80% FPR on Delve (1K) compared to 5.55% on Delve (8K) with the T5-Large architecture. In contrast, both baselines show sensitivity to the set size, with notable differences in performance, such as a 25.63% FPR on Delve (1K) compared to 18.28% on Delve (8K) using the T5-Large architecture.

**GLM Architecture Size and Type.** We investigated how the GLM architecture's size and type independently influence detection performance. Table 2 demonstrates that increasing GLM size consistently enhances outlier detection across all data domains. For example, T5-Large achieves a 5.80% FPR at 95% TPR on Delve (1K) compared to 32.30% using T5-Base. This boost can be attributed to a greater amount of parameters in T5-Large, enabling better comprehension of paragraph relationships and coherent summaries, and contributing to improved detection. Additionally, T5-Large

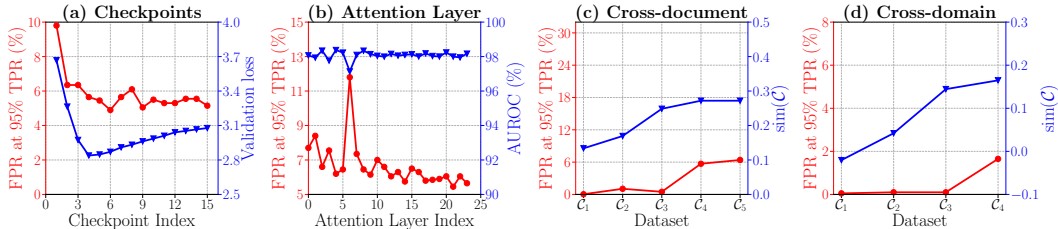

Figure 2: Performance of CODE under different settings. (a) Performance of CODE vs. pre-training validation loss under different checkpoints. (b) Performance of CODE vs. different choice of attention layers. (c) Similarities between coherent and outlier paragraphs vs. detection performance. $\mathcal{C}_1$ to $\mathcal{C}_5$ represent CNN/Daily Mail, S2orc, SAMSum, Delve (8K) and Delve (1K), respectively. (d) Performance of CODE vs. different domains. The coherent paragraphs sourced from the Delve domain, and varying outlier domains represented as $\mathcal{C}_1$ through $\mathcal{C}_4$, encompassing SAMSum, CNN/Daily Mail, Random Domain, and S2orc.

consistently outperforms BART-Large across various metrics. This highlights the critical role of architecture choice in achieving robust outlier detection in document summarization tasks. For additional insights into the performance of the BART architecture, refer to Table 9 in Appendix D.

**Pre-training Checkpoint.** We explored the impact of checkpoint selection during the pre-training phase on outlier detection. To illustrate, we tracked the summarization and detection performance of checkpoints during pre-training using the T5-Large architecture on Delve. In Figure 2 (a), we plotted pre-training validation loss against the detection FPR of CODE at each checkpoint. Our findings show that during the initial four epochs of pre-training, validation loss consistently decreases, leading to a notable reduction in detection FPR. This suggests that domain-specific pre-training enhances detection within those domains. However, as the pre-training continues, we observed an increase in validation loss, indicating potential overfitting. Intriguingly, the detection FPR remains relatively stable, implying that while overfitting may occur during pre-training, it might not significantly impact the outlier detection performance of CODE.

**Attention Layer.** In CODE, we input the output from the final cross-attention layer into the detector. Both T5 and BART architectures consist of multiple cross-attention layers, prompting us to investigate how the choice of cross-attention layers impacts detection performance, as shown in Figure 2 (b). Our findings consistently show that the lowest FPR at 95% TPR and the highest AUROC consistently occur in the cross-attention layer closest to the final layer, which is adjacent to the output layer, across all configurations. Additionally, in Figure 2 (b), we observed that the last three layers exhibit similar detection FPRs. This indicates that performance variation is minimal when selecting attention layers near the output.

**Cross-document Detection Domain.** Detection performance is notably affected by the degree of similarity between outlier and coherent paragraphs. Greater similarity between them poses a more challenging outlier detection task. To quantify this similarity, we calculated the average cosine similarity between the embeddings of coherent and outlier paragraphs within a paragraph sequence. Specifically, we employed the Sentence-BERT model (Reimers & Gurevych, 2019) to extract paragraph embeddings. The formal definition of similarity between outlier and coherent paragraphs in dataset $\mathcal{C}$ is represented as follows, where $H(X)$ denotes the embedding vector of paragraph $X$:

$$\text{sim}(\mathcal{C}) = \frac{1}{|\mathcal{C}|} \sum_{\mathcal{X} \in \mathcal{C}} \left[ \frac{1}{|\mathcal{X}^{\text{out}}|(|\mathcal{X}| - |\mathcal{X}^{\text{out}}|)} \sum_{X \in \mathcal{X}^{\text{out}}} \sum_{X' \in \mathcal{X} \setminus \mathcal{X}^{\text{out}}} \frac{\langle H(X), H(X') \rangle}{\|H(X)\|_2 \cdot \|H(X')\|_2} \right],$$

In Figure 2 (c), we depicted dataset similarity and detection performance across various domains using the T5-Large architecture. Our observations show that as outlier paragraphs become more similar to coherent ones, the detection of FPR increases. This suggests a positive correlation between the similarity of coherent and outlier paragraphs and detection errors. Additional results for other architectures can be found in Appendix G.

**Cross-domain Outlier Detection.** Table 2 presents the detection performance of CODE when coherent and outlier paragraphs are from the same domain but different documents. We anticipate this performance consistency even when fine-tuning hyper-parameters of CODE in one domain for detecting outliers in another. Figure 2 (d) depicts performance variations in diverse cross-domain outlier detection scenarios. Specifically, we pre-trained the text summarizer and selected the best hyper-

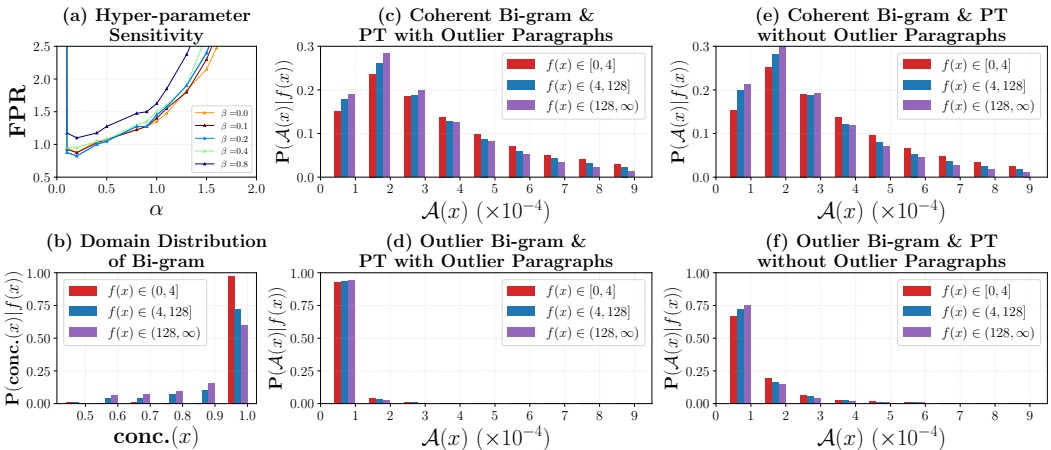

Figure 3: (a) FPR at a 95% TPR for our method under various hyper-parameters, evaluated on T5-Large and S2orc testset. (b) Domain distribution of bigrams with different occurrences. Figures (c) to (f) show bi-gram distributions. Bi-grams are from coherent paragraphs in (c) and (e) and from outlier paragraphs in (d) and (f). GLM is pre-trained with outliers in (c) and (d) and without outliers in (e) and (f). The x and y-axis represent the cross-attention $\mathcal{A}(x)$ and conditional distribution of $\mathcal{A}(x)$ under different occurrences, respectively.

parameters of CODE on the Delve domain. Next, we evaluated the detection performance against outlier paragraphs from other domains, including S2orc, Random, SAMsum and CNN Daily/Mail domains, utilizing the T5-Large architecture. Additional results for other pre-trained models are in Appendix H. In Figure 2 (d), CODE demonstrates robust performance across different domains, with a maximum false positive rate (FPR) at 95% true positive rate (TPR) of only 1.64%. We also explored the influence of the similarity between cross-domain outlier and coherent paragraphs on performance. We quantified this similarity for each cross-domain outlier detection dataset, as presented in Figure 2 (d). We observed that as the similarity between cross-domain datasets increased, the FPR at 95% TPR also increased. This implies that when coherent and outlier paragraphs closely resemble each other, the task becomes more challenging, resulting in higher detection errors.

## 6   DISCUSSIONS

In this section, we investigate the effectiveness of word frequency, cross-attention and cross-document outlier paragraphs used in the pre-training phase.

**Effectiveness of Word Frequency Hyper-parameter $\beta$.** Given the richer semantic content in bi-gram phrases compared to individual words, we use the bi-gram phrases as our primary unit of analysis. In CODE, for each bi-gram $x$ in paragraph $X$, we calculate the average attention scores with words in the summary $\hat{Y}$ and normalize it by the frequency of $x$ raised to the power $\beta$. We select a positive $\beta$ to accentuate the effects of infrequent bi-grams. Figure 3 (a) showcases how detection error varies with different $\beta$ values. Optimal results are attained with a positive $\beta$, but performance declines if $\beta$ is too large, suggesting the importance of moderate emphasis on infrequent words. To understand this, we conduct the following experiment. We determine their occurrence in four domains: CNN/Daily Mail, SAMSum, S2orc and Delve, represented as $f_1(x)$ to $f_4(x)$. The total occurrence of a phrase $x$ is $f(x) = \sum_i f_i(x)$. The metric *concentration* is defined as $\mathbf{conc.}(x) = \frac{\max_i f_i(x)}{f(x)}$, representing how bi-gram phrases are concentrated among domains. In Figure 3 (b), bi-grams with fewer than five occurrences are domain-specific, whereas those with more than 128 are domain-agnostic. Emphasizing infrequent bi-grams can enhance outlier paragraph detection since domain-specific phrases differ significantly across domains. Moreover, infrequent bi-grams typically exhibit higher average cross-attentions compared to their frequent counterparts, which may also benefit outlier detection. To see this, let $\mathcal{A}(x) = \frac{1}{|\hat{Y}|} \sum_{\hat{y} \in \hat{Y}} Att^\alpha(\hat{y}, x)$ represent the mean cross-attention between summary $\hat{Y}$ and bi-gram $x$. Figures 3 (c) and (d) display the distribution of $\mathcal{A}(x)$ for bi-grams in coherent and outlier paragraphs, respectively, across different bi-gram occurrence regimes. We observe higher average cross-attentions on less frequent bi-grams. However, this does not imply that frequent bi-grams are inconsequential in identify-

ing coherent paragraphs. Some, especially those with very high occurrence counts, may also be domain-specific terminologies. For instance, the term "Manchester United" appears 1,552 times but is exclusively found in the CNN/Daily Mail domain. Overemphasizing $\beta$ can diminish the contribution of these domain-specific terminology, potentially degrading performance. Hence, this may explain Figure 3 (a) in which as $\beta$ further increases after 0.2, the detection error increases.

**Effectiveness of Cross-Attention Hyper-parameter $\alpha$.** Comparing Figure 3 (c) and (d), we observe that the bi-grams in coherent paragraphs tend to have larger average cross-attentions than the outlier counterparts. To amplify the discrepancy between the cross-attentions of outlier and coherent bi-grams, an optimal choice of $\alpha$ is required. To see this, given the cross-attention scores of a coherent bi-gram $a_1$ and an outlier bi-gram $a_2$, with $0 < a_2 < a_1 < 1$, the difference in the powered cross-attention scores, $a_1^\alpha - a_2^\alpha$, can be maximized by selecting $\alpha^* = \frac{\ln|\ln a_1| - \ln|\ln a_2|}{\ln a_1 - \ln a_2} > 0$. The difference escalates when $\alpha < \alpha^*$ and contracts when $\alpha > \alpha^*$. This observation aligns with Figure 3 (a), where detection error initially diminishes with increasing $\alpha$ up to 0.2, and subsequently rises for all $\beta$ choices.

**Effectiveness of Outlier Paragraphs in Pre-training.** We employed the T5-Large architecture for pre-training on the Delve dataset, deliberately excluding all cross-document outlier paragraphs. Comprehensive pre-training results can be found in Appendix J. Subsequent deployment of CODE on this model yielded an 80.45% FPR at 95% TPR on the Delve outlier detection dataset. This starkly contrasts with the 5.8% FPR achieved when outliers were incorporated during pre-training. To understand the discrepancy in detection performance, we juxtapose the cross-attention distributions from Figure 3 (e) and (f) against those from Figure 3

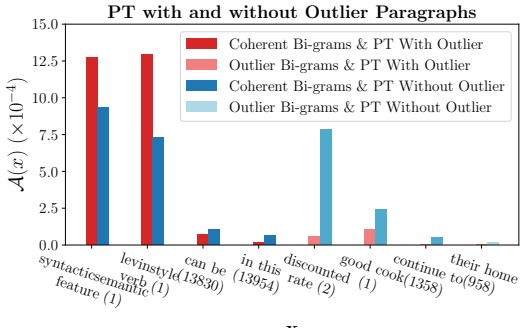

Figure 4: Cross-attention scores on eight bi-grams when T5-Large is pre-trained with and without outliers. Bi-gram occurrences are in the parenthesis.

(c) and (d). Our observations underscore that incorporating outliers during pre-training can efficaciously diminish the cross-attention scores of outlier bi-grams (i.e., comparing Figure 3 (f) to (d)), without impinging on the scores of coherent bi-grams (i.e., comparing Figure 3 (e) to (c)).

To provide more insights, we spotlight eight bi-gram phrases, of which half originate from coherent paragraphs and the remainder from outlier paragraphs. Furthermore, half of these bi-grams frequently appear, as indicated by their occurrence counts in parenthesis. Comparing the cross-attention scores when the T5-Large model is pre-trained with (i.e., red bars) and without (i.e., blue bars) outliers, we observed that including outliers enhances the attention scores of less frequent bi-grams in coherent paragraphs, simultaneously depressing scores for the less frequent outlier bi-grams. For instance, after incorporating outliers in pre-training, the coherent bi-gram "levinstyle verb" with a single occurrence nearly doubles its attention score, whereas the outlier bi-gram "discounted rate" with two occurrences sees an 80% attention reduction. Moreover, we observed that the attention scores of domain-agnostic phrases also wane, potentially bolstering outlier detection capabilities. For example, after incorporating outliers in pre-training, we observe notable reductions in attention scores for the domain-agnostic phrases "can be" in coherent paragraphs and "continue to" in outlier paragraphs.

## 7 CONCLUSIONS

In conclusion, while generative text summarizers excel at content representation, their vulnerability to outlier paragraphs poses significant challenges. In this paper, we focus on cross-document and cross-domain outlier paragraph detection. By exploiting the encoder-decoder cross-attention structure and unique behaviors of infrequent words, we introduced CODE, a novel and efficient outlier detector. Experimental results validate the superiority of CODE over the traditional supervised fine-tuning methods under various datasets and architectures. Our findings illuminate the potential of harnessing cross-attention distribution, word frequency nuances and the strategic use of cross-document outliers in the pre-training phase, setting a promising direction for future advancements in the realm of text summarization.

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

# A    RELATED WORK

The challenge of detecting out-of-distribution (OOD) samples by deep neural networks has received extensive research attention (Sun et al., 2022; Lakshminarayanan et al., 2017; Hendrycks et al., 2018; Yang et al., 2021; Fort et al., 2021). A class of methods used to solve the OOD problem is softmax-based methods (Hendrycks & Gimpel, 2016; Liang et al., 2017; Hsu et al., 2020). As an initial attempt, Hendrycks & Gimpel (2016) leverages the softmax probability of the model's output to identify OOD samples, effectively mitigating the issue of neural networks assigning excessively high confidence scores to such data (Nguyen et al., 2015). A prominent representative of softmax-based methods is ODIN (Liang et al., 2017), which incorporates temperature scaling and input perturbations to significantly enhance the efficiency of both in-distribution and out-of-distribution sample detection. Lee et al. (2018) opts for a different path by computing confidence scores for input samples using the Mahalanobis distance (Mahalanobis, 2018) instead of relying on softmax. Notably, this method fine-tunes the model using OOD data, and similar settings are used in (Vyas et al., 2018). Diverging from the conventional softmax-based methodology, our approach utilizes the cross-attention matrix within the generative language model for the purpose of detecting outlier paragraphs. Importantly, our method retains the original model structure and eliminates the need for additional fine-tuning. In addition, our method is to implement outlier paragraph detection inside the sample instead of OOD sample detection. Our detection objective aligns with the concept of anomaly detection using generative adversarial networks (Schlegl et al., 2017; 2019), albeit without the extensive model fine-tuning typically associated with such endeavors.

# B    SUPPLEMENTARY MATERIALS FOR SECTION 4.2 AND SECTION 4.3

## B.1    DETAILED DONSTRUCTION METHOD OF EACH PRE-TRAINING DATASET

In this subsection, we introduce the construction details of pre-training datasets CNN/Daily Mail-PT, SAMSum-PT, Delve-PT, and S2orc-PT in detail.

**CNN/Daily Mail-PT.** For the limitation of model input length, we use samples whose source document length is less than five hundred words as samples to be injected. We split the source document in these samples into two coherent paragraphs. We split the source documents in the remaining samples into multiple paragraphs and collected them as candidate outlier paragraphs. For each sample to be injected, we randomly select two outlier paragraphs to insert.

**SAMSum-PT.** We divide the dataset into two parts at a ratio of 1:1, one part is prepared to be injected and the other part is used to provide outlier paragraphs. For the samples to be inserted, we also split the source document into two paragraphs. We split the input document in another part of the samples into two paragraphs. We collect these paragraphs as candidate outlier paragraphs. For each sample to be injected, we randomly select two outlier paragraphs for insertion.

**Delve-PT and S2orc-PT.** We view the citation markers in the summary paragraphs to find coherent paragraphs and outlier paragraphs. Specifically, we select summaries with at least two citation markers. We randomly select two markers when a summary contains multiple citation markers. Next, for each citation marker in a summary, we find the corresponding paper abstracts as coherent paragraphs. To get outlier paragraphs, we use Microsoft Academic Graph (MAG) (Shen et al., 2018) to determine the academic fields where the abstract belongs. For each abstract, MAG directly provides their academic fields in a hierarchical manner with a progressively finer granularity from L0 to L5. To get the outlier paragraphs, under L3 and more specific sub-fields, we select abstract paragraphs whose fields do not intersect with coherent abstract paragraphs. We also insert two outlier paragraphs into each sample.

## B.2    ADDITIONAL DATASET STATISTICS

In this subsection, we report the statistics of the pre-training datasets, the cross-document outlier paragraph detection dataset, and the test sets of cross-domain outlier paragraph detection. These statistics are presented in Tables 3, 4 and 5, respectively.

Table 3: Additional statistics of the pre-training datasets with outlier paragraphs.

|  | | # Examples | # Words (single) | # Words (all) |
|---|---|---|---|---|
| **CNN/Daily Mail** | Paragraph | 202,220 | avg: 223.19, std: 55.08 | 361,363 |
|  | Summary | 52,459 | avg: 47.78, std: 21.13 | 85,486 |
| **SAMSum** | Paragraph | 13,180 | avg: 61.35, std: 48.36 | 22,005 |
|  | Summary | 4,092 | avg: 23.53, std: 12.75 | 8,731 |
| **Delve** | Paragraph | 34,159 | avg: 173.66, std: 104.07 | 97,055 |
|  | Summary | 10,000 | avg: 30.82, std: 15.71 | 19,667 |
| **S2orc** | Paragraph | 85,444 | avg: 217.12, std: 172.381 | 182,478 |
|  | Summary | 24,000 | avg: 34.72, std: 18.64 | 42,019 |

Table 4: Additional statistics of the cross-document outlier paragraph detection datasets.

|  | | # Examples | # Words (single) | # Words (all) |
|---|---|---|---|---|
| **CNN/Daily Mail** | Paragraph | 98,221 | avg: 220.50, std: 55.42 | 239,549 |
| **SAMSum** | Paragraph | 13,186 | avg: 61.95, std: 50.00 | 22,349 |
| **Delve** | Paragraph | 35,039 | avg: 173.32, std: 91.25 | 98,778 |
| **S2orc** | Paragraph | 16,167 | avg: 216.81, std: 177.46 | 74,100 |

Table 5: Additional statistics of the cross-domain outlier paragraph detection test sets. A ← B means sampling the outlier paragraphs from dataset B and inserting them into dataset A.

|  | | # Examples | # Words (single) | # Words (all) |
|---|---|---|---|---|
| **CNN/Daily Mail ← SAMSum** | Paragraph | 5,495 | avg: 185.32, std: 78.76 | 45,360 |
| **CNN/Daily Mail ← Delve** | Paragraph | 6,817 | avg: 191.72, std: 79.56 | 55,659 |
| **CNN/Daily Mail ← S2orc** | Paragraph | 7,816 | avg: 203.37, std: 111.46 | 63,996 |
| **CNN/Daily Mail ← Random domain** | Paragraph | 8,931 | avg: 177.61, std: 60.25 | 279,440 |
| **SAMSum ← CNN/Daily Mail** | Paragraph | 1,581 | avg: 150.11, std: 97.94 | 20,845 |
| **SAMSum ← Delve** | Paragraph | 1,488 | avg: 110.57, std: 87.16 | 13,662 |
| **SAMSum ← S2orc** | Paragraph | 1,541 | avg: 137.78, std: 154.99 | 17,940 |
| **SAMSum ← Random domain** | Paragraph | 1,607 | avg: 105.78, std: 59.43 | 99,622 |
| **Delve ← CNN/Daily Mail** | Paragraph | 3,538 | avg: 201.81, std: 68.32 | 38,924 |
| **Delve ← SAMSum** | Paragraph | 2,405 | avg: 143.66, std: 81.65 | 18,064 |
| **Delve ← S2orc** | Paragraph | 3,468 | avg: 184.48, std: 111.51 | 30,023 |
| **Delve ← Random domain** | Paragraph | 3,694 | avg: 158.70, std: 57.67 | 185,256 |
| **S2orc ← CNN/Daily Mail** | Paragraph | 6,571 | avg: 232.75, std: 160.62 | 61,800 |
| **S2orc ← SAMSum** | Paragraph | 2,405 | avg: 143.66, std: 81.65 | 18,064 |
| **S2orc ← Delve** | Paragraph | 5,546 | avg: 209.02, std: 184.34 | 41,473 |
| **S2orc ← Random domain** | Paragraph | 7,075 | avg: 190.919, std: 159.76 | 255,651 |

# C   SUPPLEMENTARY MATERIALS FOR EXPERIMENTAL SETUPS

## C.1   PRE-TRAINING SETUPS

In this subsection, we report the pre-training hyper-parameter settings in Table 6.

Table 6: Pre-training settings of the GLMs. Characters "B" and "L" denote the model size of Base and Large, respectively. All models are trained on the Tesla A100 machine. We set warm-up steps to 200 and employ a linear learning rate scheduler.

| Datasets | Models | Learning rate | # Epochs | Batch size |
|---|---|---|---|---|
| CNN/Daily Mail-PT | BART-B | 0.00003 | 15 | 8 |
| | BART-L | 0.00003 | 15 | 4 |
| SAMSum-PT | BART-B | 0.00003 | 15 | 8 |
| | BART-L | 0.00003 | 15 | 4 |
| Delve-PT | BART-B | 0.00003 | 15 | 16 |
| | BART-L | 0.00003 | 15 | 8 |
| S2orc-PT | BART-B | 0.00003 | 15 | 8 |
| | BART-L | 0.00003 | 15 | 8 |
| CNN/Daily Mail-PT | T5-B | 0.0002 | 15 | 6 |
| | T5-L | 0.0001 | 15 | 6 |
| SAMSum-PT | T5-B | 0.0002 | 15 | 6 |
| | T5-L | 0.0001 | 15 | 6 |
| Delve-PT | T5-B | 0.0002 | 15 | 6 |
| | T5-L | 0.0001 | 15 | 6 |
| S2orc-PT | T5-B | 0.0002 | 15 | 12 |
| | T5-L | 0.0001 | 15 | 6 |

## C.2 Performance of the Pre-trained Models

In this subsection, we show the performance of text summarization on each dataset and pre-trained model in Table 7. We use ROUGE to evaluate the quality of text summarization and performance of all pre-trained models.

Additionally, the metrics used in this section are as follows:

- **ROUGE-1** measures the overlap of unigrams between the reference and the generated summary.
- **ROUGE-2** extends the concept of ROUGE-1 to bigrams, measuring the overlap of consecutive pairs of words between the reference and the generated summary.
- **ROUGE-L** calculates the longest common subsequence between the reference and the generated summary.

Table 7: Performance of the pre-trained models

| Datasets | Models | ROUGE-1 | ROUGE-1 | ROUGE-L |
|---|---|---|---|---|
| Delve | T5-L | 19.3443 | 3.3781 | 14.4185 |
| | T5-B | 17.5721 | 2.8855 | 13.4359 |
| | BART-L | 18.0474 | 2.7043 | 13.6427 |
| | BART-B | 18.3348 | 2.8605 | 13.9695 |
| S2orc | T5-L | 20.4524 | 3.9853 | 15.1929 |
| | T5-B | 19.9058 | 3.6515 | 14.7904 |
| | BART-L | 20.7972 | 3.7129 | 15.4441 |
| | BART-B | 19.9070 | 3.4996 | 14.8250 |
| SAMsum | T5-L | 44.3738 | 21.7557 | 38.7138 |
| | T5-B | 43.1620 | 20.6720 | 38.6918 |
| | BART-L | 50.4676 | 25.7701 | 41.8661 |
| | BART-B | 44.9713 | 20.4162 | 36.2211 |
| CNN/Daily Mail | T5-L | 35.5728 | 12.0295 | 25.0173 |
| | T5-B | 33.7640 | 14.7571 | 23.3762 |
| | BART-L | 41.8007 | 20.1378 | 30.1265 |
| | BART-B | 41.4113 | 19.7040 | 29.7622 |

We also note here that on the CNN/Daily Mail dataset, the reference Lewis et al. (2019) reports 44.16, 21.28, and 40.90 on the BART model, and the reference Raffel et al. (2020) reports 43.52, 21.55 and 40.69 on T5 model, respectively. Our pre-trained model generally has worse performance, since (1) we add the outlier paragraphs in the pre-trained phrase; (2) the data for outlier paragraph detection is constructed from the raw data, and some part of the training set from Lo et al. (2019)Akujuobi & Zhang (2017)Gliwa et al. (2019)Nallapati et al. (2016) are used for hyper-parameter search. Therefore, the total amount of training data is smaller than the original dataset, which may lead to a worse performance of text summarization. Although the performance of our pre-training model is worse, this does not affect the effectiveness of outlier paragraph detection.

## C.3 Training Setups of the Baselines

In this subsection, we report the training settings of the Frozen and FT-ALL. Table 8 presents the training epochs and batch sizes.

**Frozen.** We use the AdamW optimizer with exponential decay rates for the first and second moments of the gradient updates setting to 0.9 and 0.999, respectively. We choose a constant learning rate scheduler with a warm-up period of 200 steps. The learning rates are selected from the set $\{10^{-6}, 10^{-5}, 10^{-4}, 10^{-3}\}$. The weight decay parameter is configured to be 0.0001. For each hyper-parameter setting, we run three times with different random seeds. In the main paper, we report the

mean value of the results, while the standard deviations are presented in Table 10. We select the model with the lowest validation loss for testing in outlier paragraph detection.

**FT-ALL.** We utilize the same hyper-parameter setting used in the baseline Frozen, except that the learning rate is set to the one used in the summarizer pre-training. We repeat this baseline three times with different random seeds.

Table 8: Epochs and batch size of the baselines. Characters "B" and "L" denote the model size of Base and Large, respectively. All models are trained on the Tesla A100 machine.

|  | Datasets | Models | # Epochs | Batch size |
|---|---|---|---|---|
| **Frozen** | CNN/Daily Mail | BART-B | 40 | 64 |
|  |  | BART-L | 40 | 64 |
|  | SAMSum | BART-B | 40 | 64 |
|  |  | BART-L | 40 | 64 |
|  | Delve (1K) | BART-B | 40 | 64 |
|  |  | BART-L | 40 | 64 |
|  | Delve (8K) | BART-B | 40 | 64 |
|  |  | BART-L | 40 | 64 |
|  | S2orc | BART-B | 40 | 64 |
|  |  | BART-L | 40 | 64 |
|  | CNN/Daily Mail | T5-B | 40 | 64 |
|  |  | T5-L | 40 | 64 |
|  | SAMSum | T5-B | 40 | 64 |
|  |  | T5-L | 40 | 64 |
|  | Delve (1K) | T5-B | 40 | 64 |
|  |  | T5-L | 40 | 64 |
|  | Delve (8K) | T5-B | 40 | 64 |
|  |  | T5-L | 40 | 64 |
|  | S2orc | T5-B | 40 | 64 |
|  |  | T5-L | 40 | 64 |
| **FT-ALL** | CNN/Daily Mail | BART-B | 10 | 8 |
|  |  | BART-L | 10 | 8 |
|  | SAMSum | BART-B | 10 | 8 |
|  |  | BART-L | 10 | 8 |
|  | Delve (1K) | BART-B | 10 | 8 |
|  |  | BART-L | 10 | 8 |
|  | Delve (8K) | BART-B | 10 | 8 |
|  |  | BART-L | 10 | 8 |
|  | S2orc | BART-B | 10 | 8 |
|  |  | BART-L | 10 | 8 |
|  | CNN/Daily Mail | T5-B | 10 | 8 |
|  |  | T5-L | 10 | 8 |
|  | SAMSum | T5-B | 10 | 8 |
|  |  | T5-L | 10 | 8 |
|  | Delve (1K) | T5-B | 10 | 4 |
|  |  | T5-L | 10 | 4 |
|  | Delve (8K) | T5-B | 10 | 4 |
|  |  | T5-L | 10 | 4 |
|  | S2orc | T5-B | 10 | 4 |
|  |  | T5-L | 10 | 4 |

# D    Supplementary Results in Main Results

In this section, we present all evaluation results to show the improvement of our method compared to the baselines. Table 9 shows the performance of our proposed method and two baselines under each dataset. The details of our method and the baselines can be found in section 3. We note here that our method is deterministic and does not have an error bar. The other two baselines are randomly re-initialized with three different seeds. We take the average of the results as the final performance and calculate the standard deviation. Table 10 provides the standard deviation for different models. Table 11 provides the hyper-parameters $\alpha$ and $\beta$ of CODE are used in the evaluation process.

The evaluation metrics used in section 5 are as follows:

- **FPR at 95% TPR** refers to the rate that an outlier paragraph is misclassified as a coherent paragraph when the true positive rate (TPR) is at 95%.

- **AUROC** is calculated as the Area Under the Receiver Operating Characteristic curve (Fawcett, 2006). The ROC curve illustrates the relationship between TPR and FPR at various thresholds. The higher the value of AUROC, the stronger the discriminative ability of the model.

- **AUPR** stands for Area Under the Precision-Recall curve (Manning & Schutze, 1999; Saito & Rehmsmeier, 2015). The PR curve depicts the trade-off between precision and recall at various thresholds. For an ideal classifier, its AUPR score is 1.

Table 9: Evaluation results of CODE and baselines for cross-document outlier paragraph detection. ↑ indicates that larger values are better, and ↓ indicates that smaller values are better. Characters "B" and "L" denote the Base and Large model, respectively.

| | Models | FPR (95%) TPR ↓ | AUROC ↑ | AUPR ↑ |
|---|---|---|---|---|
| | | **CODE/Frozen/FT-ALL** | | |
| Delve (1K) | T5-L | **5.80**/30.30/25.63 | **98.08**/92.87/94.59 | **97.03**/93.57/92.60 |
| | T5-B | **32.30**/65.97/57.75 | **90.08**/84.52/85.21 | **83.76**/82.62/82.92 |
| | BART-L | **11.10**/43.02/44.45 | **96.09**/91.23/91.84 | **93.41**/90.08/90.47 |
| | BART-B | **19.65**/49.27/53.02 | **91.60**/90.62/90.99 | **93.66**/90.23/90.61 |
| Delve (8K) | T5-L | **5.55**/16.85/18.28 | **98.16**/93.62/95.87 | **97.23**/94.01/95.18 |
| | T5-B | **31.50**/60.22/47.98 | **90.36**/86.32/87.64 | **84.34**/85.40/87.49 |
| | BART-L | **11.10**/33.52/33.45 | **96.09**/93.17/92.75 | **93.41**/92.96/91.61 |
| | BART-B | **20.30**/45.40/38.00 | **94.79**/90.66/92.04 | **91.30**/89.98/90.95 |
| S2orc | T5-L | **1.08**/10.40/6.05 | **99.54**/96.01/97.69 | **99.27**/95.59/97.32 |
| | T5-B | **2.53**/15.82/11.65 | **99.00**/96.68/96.87 | **97.95**/96.51/96.01 |
| | BART-L | **4.83**/16.18/9.47 | **98.66**/96.03/96.77 | **98.11**/95.45/96.15 |
| | BART-B | **3.00**/6.94/5.07 | **98.72**/97.91/97.71 | **97.56**/97.55/97.26 |
| SAMsum | T5-L | **0.60**/5.50/0.65 | **99.87**/98.67/99.68 | **99.87**/98.78/98.60 |
| | T5-B | **0.61**/8.44/1.22 | **99.66**/99.21/97.46 | **99.43**/99.00/96.68 |
| | BART-L | **0.91**/0.65/0.28 | **99.43**/99.70/99.77 | **99.37**/99.67/99.77 |
| | BART-B | **2.26**/3.83/3.67 | **97.23**/99.15/97.83 | **94.61**/99.18/97.83 |
| CNN/Daily Mail | T5-L | **0.00**/0.20/0.32 | **99.99**/99.85/99.77 | **99.99**/99.81/99.79 |
| | T5-B | **0.12**/0.82/0.29 | **99.96**/99.62/99.80 | **99.96**/99.56/99.70 |
| | BART-L | **0.14**/0.57/0.44 | **99.71**/99.69/99.78 | **99.60**/99.73/99.75 |
| | BART-B | **0.18**/0.23/0.33 | **99.89**/99.87/99.86 | **99.83**/99.86/99.86 |

Table 10: Standard deviation of the evaluation results.

| | Models | FPR (95%) TPR ↓ | AUROC ↑ | AUPR ↑ |
|---|---|---|---|---|
| | | **CODE/Frozen/FT** | | |
| Delve (1K) | T5-L | 0.00 /0.94/1.34 | 0.00/0.21/0.91 | 0.00/0.16/0.76 |
| | T5-B | 0.00/1.53/7.42 | 0.00/0.20/9.83 | 0.00/0.16/12.46 |
| | BART-L | 0.00/1.17/2.49 | 0.00/0.19/0.39 | 0.00/0.20/0.40 |
| | BART-B | 0.00/1.42/0.34 | 0.00/0.13/0.06 | 0.00/0.21/0.10 |
| Delve (8K) | T5-L | 0.00/0.62/1.05 | 0.00/0.09/0.08 | 0.00/0.11/0.34 |
| | T5-B | 0.00/1.08/0.55 | 0.00/0.13/1.12 | 0.00/0.15/0.92 |
| | BART-L | 0.00/0.98/2.45 | 0.00/0.02/0.24 | 0.00/0.03/0.40 |
| | BART-B | 0.00/1.18/0.76 | 0.00/0.45/0.20 | 0.00/0.62/0.34 |
| S2orc | T5-L | 0.00/0.35/0.31 | 0.00/0.27/0.93 | 0.00/0.33/0.86 |
| | T5-B | 0.00/0.48/0.35 | 0.00/0.11/3.02 | 0.00/0.48/4.93 |
| | BART-L | 0.00/0.01/1.04 | 0.00/0.01/0.11 | 0.00/0.01/0.13 |
| | BART-B | 0.00/0.23/0.25 | 0.00/0.01/0.25 | 0.00/0.01/0.64 |
| SAMsum | T5-L | 0.00/0.46/0.24 | 0.00/0.03/0.01 | 0.00/0.04/0.02 |
| | T5-B | 0.00/0.43/0.32 | 0.00/0.02/0.01 | 0.00/0.03/0.03 |
| | BART-L | 0.00/0.11/0.06 | 0.00/0.01/0.02 | 0.00/0.01/0.01 |
| | BART-B | 0.00/0.12/0.46 | 0.00/0.05/0.05 | 0.00/0.06/0.21 |
| CNN/Daily Mail | T5-L | 0.00/0.01/0.00 | 0.00/0.00/0.00 | 0.00/0.02/0.00 |
| | T5-B | 0.00/0.01/0.01 | 0.00/0.01/0.00 | 0.00/0.00/0.01 |
| | BART-L | 0.00/0.06/0.10 | 0.00/0.01/0.02 | 0.00/0.01/0.01 |
| | BART-B | 0.00/0.02/0.46 | 0.00/0.01/0.05 | 0.00/0.01/0.21 |

Table 11: The hyper-parameters $\alpha$ and $\beta$ of CODE are used in the main results. Characters "B" and "L" denote the model size of Base and Large, respectively.

| | BART-B | BART-L | T5-B | T5-L |
|---|---|---|---|---|
| | **$\alpha, \beta$** | | | |
| CNN/Daily Mail-OD | 0.2, 0.0 | 0.2, 0.3 | 0.2, 0.1 | 0.2, 0.1 |
| SAMSum-OD | 0.2, 0.0 | 0.2, 0.0 | 0.4, 0.2 | 0.4, 0.4 |
| Delve-OD (1K) | 1.2, 0.2 | 0.2, 0.1 | 1.2, 0.0 | 0.2, 0.0 |
| Delve-OD (8K) | 0.8, 0.0 | 1.0, 0.1 | 1.0, 0.2 | 0.6, 0.1 |
| S2orc-OD | 0.6, 0.1 | 1.0, 0.1 | 0.6, 0.0 | 0.4, 0.0 |

# E PERFORMANCE VS. PRE-TRAINED MODEL CHECKPOINTS

In this section, we show how the selection of checkpoints of the pre-trained model affects the detection performance of our method. Specifically, we present the relationship between the validation loss for each checkpoint on the pre-trained dataset and their cross-document outlier paragraph detection performance. Each figure in this section displays the validation loss and FPR at 95% TPR metric of each dataset and model at different checkpoints. We find out that the pre-trained model with the smallest validation loss is generally not the pre-trained model with the best outlier detection performance, but the detection performance difference between the pre-trained model with the smallest validation loss and the pre-trained model with the best outlier paragraph detection performance is negligible.

The correspondence between the figures and the setting is as follows:

- Figure 5: performance on Delve (1K) dataset and four models.
- Figure 6: performance on Delve (8K) dataset and four models.
- Figure 7: performance on S2orc dataset and four models.
- Figure 8: performance on SAMsum dataset and four models.
- Figure 9: performance on CNN/Daily Mail dataset and four models.

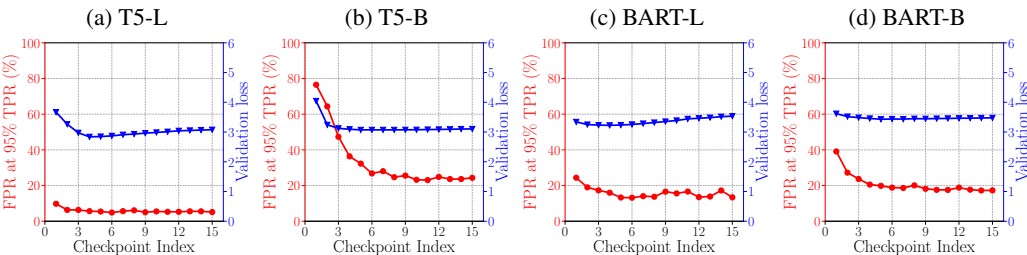

Figure 5: Performance vs. Checkpoints on Delve (1K)

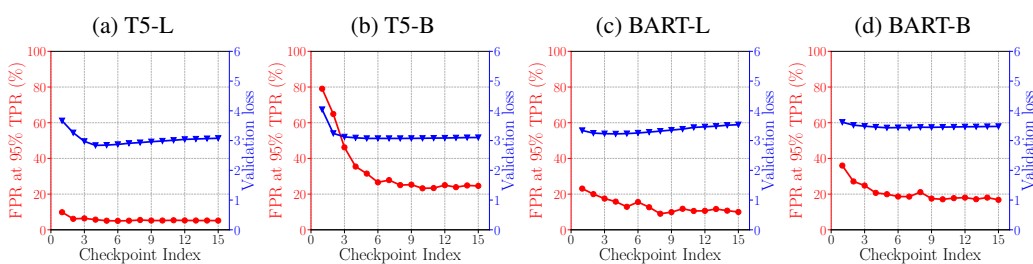

Figure 6: Performance vs. Checkpoints on Delve (8K).

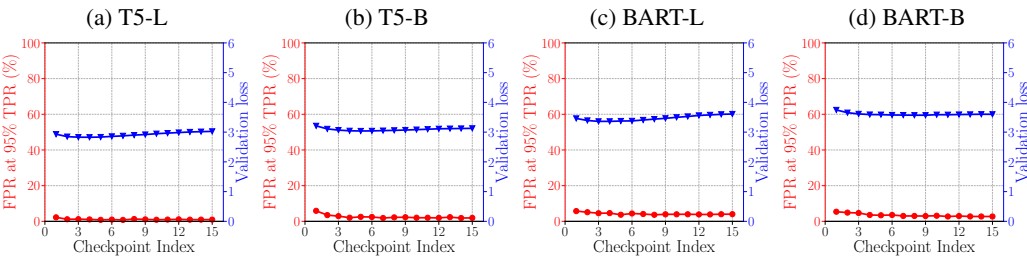

Figure 7: Performance vs. Checkpoints on S2orc

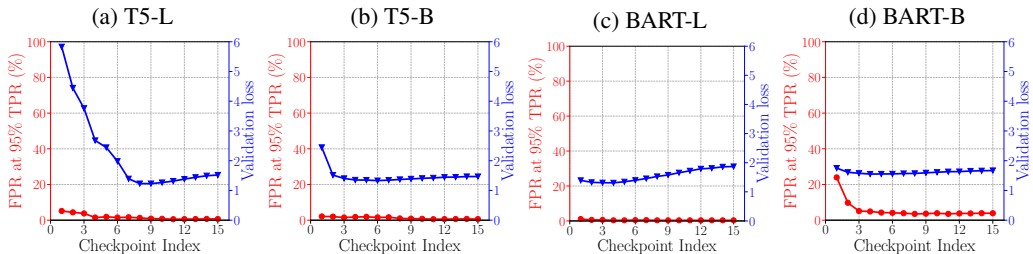

Figure 8: Performance vs. Checkpoints on SAMsum

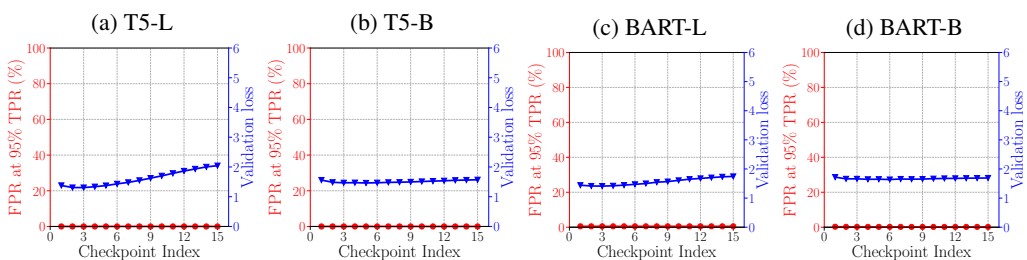

Figure 9: Performance vs. Checkpoints on CNN/Daily Mail

# F PERFORMANCE VS. PRE-TRAINED MODEL ATTENTION LAYERS

In this section, we show how different attention layers affect the outlier detection performance of our method. Specifically, we present the relationship between the attention layer and two evaluation metrics of outlier paragraph detection. Each figure in this section displays FPR at 95% TPR and AUROC of our method on each dataset and model when different attention layers are selected. We observe that the lowest FPR at 95% TPR and the highest AUROC occur in the attention layer close to the last layer (the layer closest to the output layer) for most types of models and datasets, except BART-base, which contains only six attention layers. In fact, we can also observe that the last three layers have similar performance and this indicates that the performance varies small if the attention layers close to the output layer are selected.

The correspondence between the figures and the setting is as follows:

- Figure 10: performance on Delve (1K) dataset and each model.
- Figure 11: performance on Delve (8K) dataset and each model.
- Figure 12: performance on S2orc dataset and each model.
- Figure 13: performance on SAMsum dataset and each model.
- Figure 14: performance on CNN/Daily Mail dataset and each model.

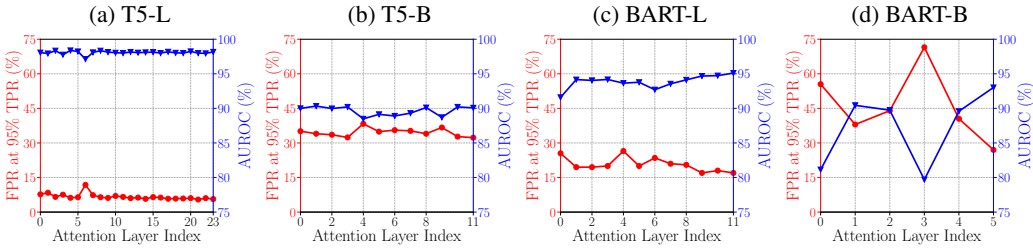

Figure 10: Performance vs. Attention Layers on Delve (1K)

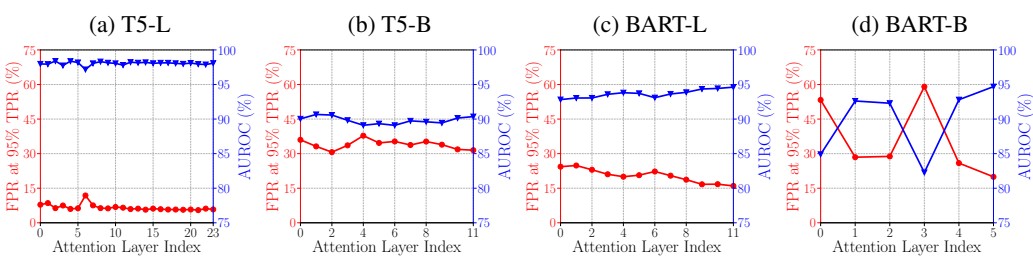

Figure 11: Performance vs. Attention Layers on Delve (8K)

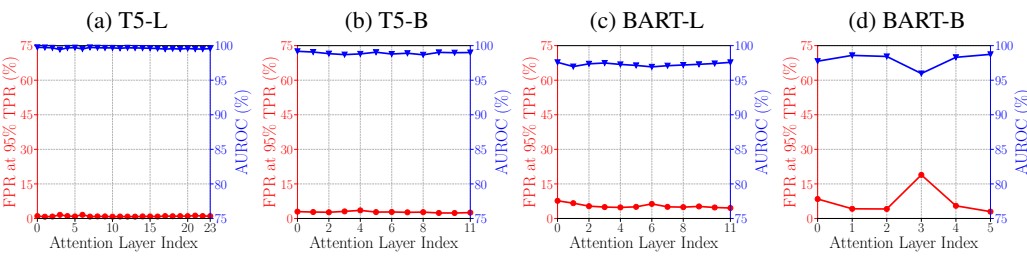

Figure 12: Performance vs. Attention Layers on S2orc

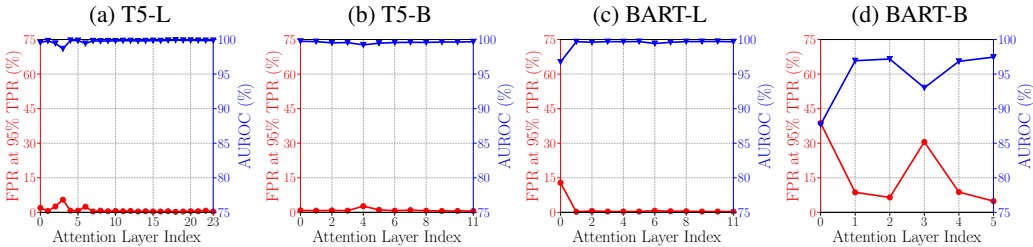

Figure 13: Performance vs. Attention Layers on SAMsum

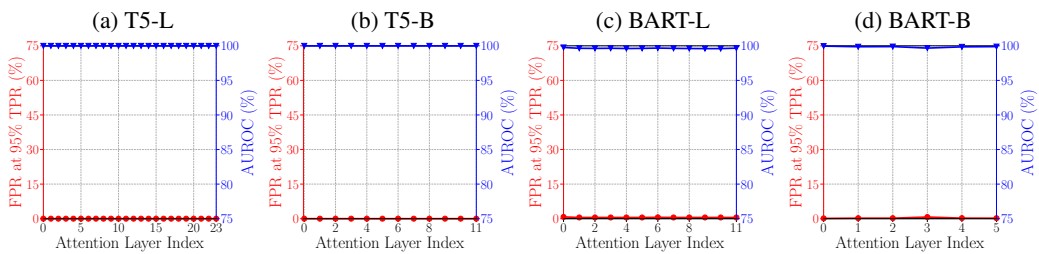

Figure 14: Performance vs. Attention Layers on CNN/Daily Mail

## G PERFORMANCE VS. CROSS-DOCUMENT OUTLIER DETECTION DIFFICULTY

In this section, we show how different dataset affects the cross-document outlier detection performance of our method. We present the relationship between the dataset similarity and two evaluation metrics of outlier paragraph detection. Figure 15 displays how FPR at 95% TPR changes with the improvement of dataset similarity, while Figure 16 displays how AUROC changes with the improvement of dataset difficulty. $\mathcal{C}_1$ to $\mathcal{C}_5$ represent CNN/Daily Mail, S2orc, SAMSum, Delve (8K), and Delve (1K), respectively.

To measure the similarity of the dataset, we use the Sentence-BERT model to obtain the embedding of input documents and calculate the average cosine similarity between the embedding of coherent and outlier paragraphs within a single data. Specifically, each paragraph contains two coherent paragraphs and two outlier paragraphs. For each paragraph $X$ in the dataset $\mathcal{C}$, we use $H(X)$ to denote the embedding vector of paragraph $X$. Therefore, the difficulty of the dataset $\mathcal{C}$ is defined as:

$$\text{sim}(\mathcal{C}) = \frac{1}{|\mathcal{C}|} \sum_{\mathcal{X} \in \mathcal{C}} \left[ \frac{1}{|\mathcal{X}^{\text{out}}|(|\mathcal{X}| - |\mathcal{X}^{\text{out}}|)} \sum_{X \in \mathcal{X}^{\text{out}}} \sum_{X' \in \mathcal{X} \setminus \mathcal{X}^{\text{out}}} \frac{\langle H(X), H(X') \rangle}{\|H(X)\|_2 \cdot \|H(X')\|_2} \right] \quad (3)$$

The higher the cosine similarity, the smaller the difference between coherent and outlier paragraphs in the dataset, indicating it is harder to detect outliers on this dataset. We observe that when the coherent and outlier paragraphs in the dataset tend to be less similar to each other (i.e., the similarity of the dataset is smaller), our method tends to have a smaller FPR at 95% TPR and a larger AUROC.

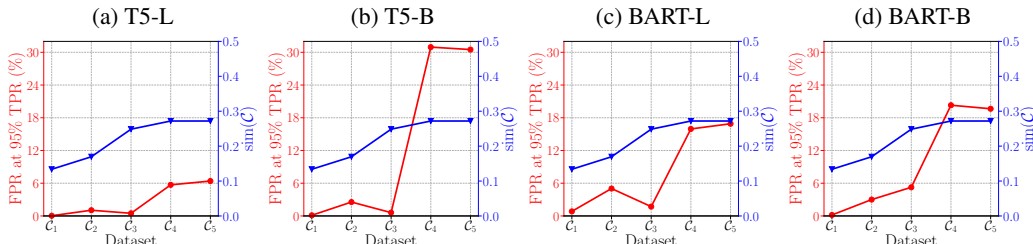

Figure 15: FPR at 95% TPR vs. $\text{sim}(\mathcal{C})$ in cross-document outlier detection.

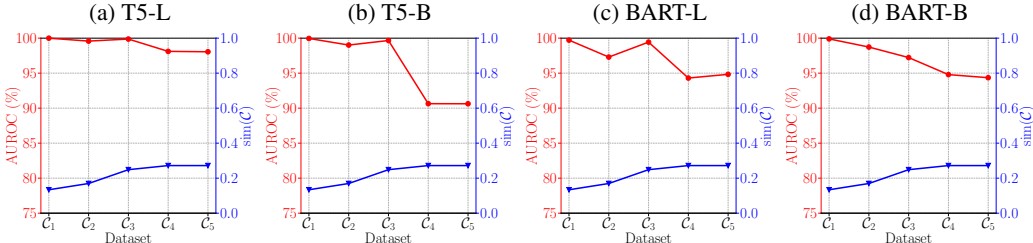

Figure 16: AUROC vs. $\text{sim}(\mathcal{C})$ in cross-document outlier detection.

# H PERFORMANCE VS. CROSS-DOMAIN OUTLIER DETECTION DIFFICULTY

In this section, we show how our method transfers across different domains. Recall that we pre-train the generative language model, find the best hyper-parameter setting, and test the detection performance on the same domain. We hope that this pre-trained model together with the best hyper-parameter setting can also transfer to other domains. Therefore, we constructed cross-domain test sets to evaluate the cross-domain performance. The details of the cross-domain dataset can be found in section 4.3, B.2, and we use equation 3 to measure the difficulty of cross-domain datasets.

We present the relationship between cross-domain dataset similarity and two evaluation metrics of the outlier paragraph detection. Figure 17, 18, 19, 20 display FPR at 95% TPR, while figure Figure 21, 22, 23, 24 display AUROC on each model and dataset.

From the figures, we observe that for most settings, FPR at 95% TPR decreases, and AUROC increases as the similarity of the dataset increases, except for one case. In Figure 18d, we observe although the s2orc-random domain has a smaller difficulty, FPR is two times larger than that of S2orc ← Delve domain. The performance on the AUROC metric is also worse than that of S2orc ← Delve domain in Figure22d. We generally observe this on the smaller model, i.e., BART-Base, consisting of nearly 140M parameters. On the larger model, we do not observe this. This may be due to the fact that the large model models tend to perform better for cross-domain data. We also observe that T5 model generally performs better than BART on most cross-domain datasets. We also observe that the larger models yield better performance for both BART and T5.

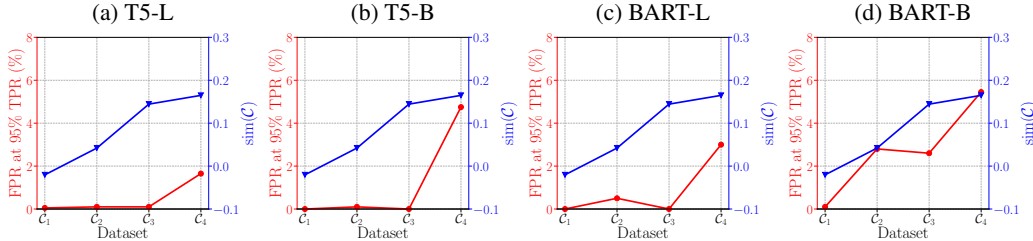

Figure 17: FPR at 95% TPR vs. $\text{sim}(\mathcal{C})$; The coherent paragraphs sourced from the Delve (1K) domain, and varying outlier domains represented as $\mathcal{C}_1$ through $\mathcal{C}_4$, encompassing SAMSum, CNN/Daily Mail, Random Domain, and S2orc.

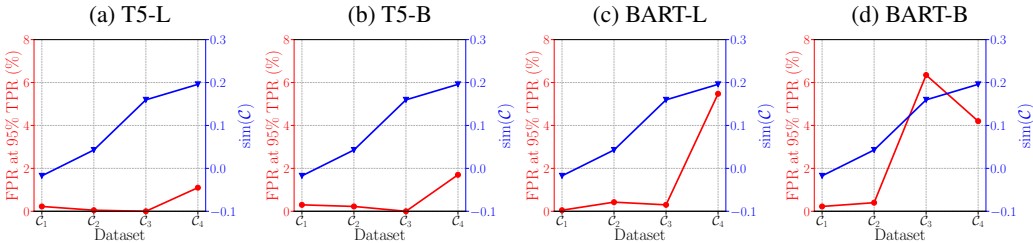

Figure 18: FPR at 95% TPR vs. $\text{sim}(\mathcal{C})$; The coherent paragraphs sourced from the S2orc domain, and varying outlier domains represented as $\mathcal{C}_1$ through $\mathcal{C}_4$, encompassing SAMSum, CNN/Daily Mail, Random Domain, and Delve.

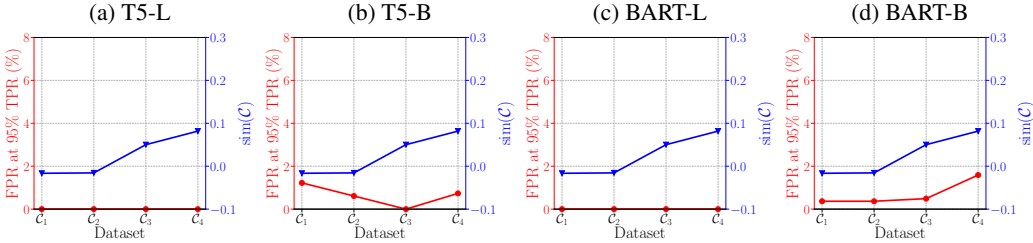

Figure 19: FPR at 95% TPR vs. $\text{sim}(\mathcal{C})$; The coherent paragraphs sourced from the SAMSum domain, and varying outlier domains represented as $\mathcal{C}_1$ through $\mathcal{C}_4$, encompassing Delve, S2orc, Random Domain, and CNN/Daily Mail.

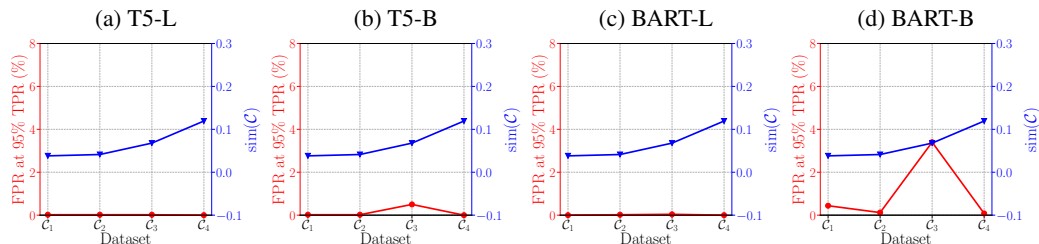

Figure 20: FPR at 95% TPR vs. $\text{sim}(\mathcal{C})$; The coherent paragraphs sourced from the CNN/Daily Mail domain, and varying outlier domains represented as $\mathcal{C}_1$ through $\mathcal{C}_4$, encompassing Delve, S2orc, SAMSum, and Random Domain.

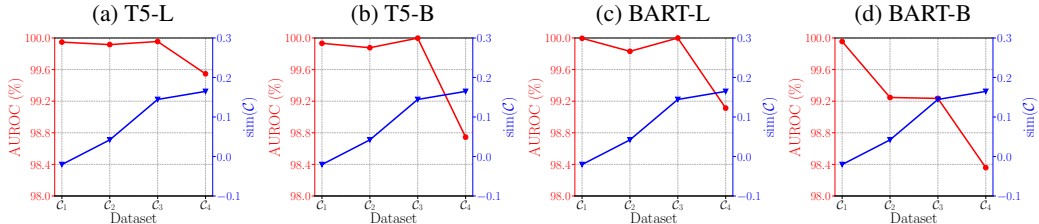

Figure 21: AUROC vs. $\text{sim}(\mathcal{C})$; The coherent paragraphs sourced from the Delve (1K) domain, and varying outlier domains represented as $\mathcal{C}_1$ through $\mathcal{C}_4$, encompassing SAMSum, CNN/Daily Mail, Random Domain, and S2orc.

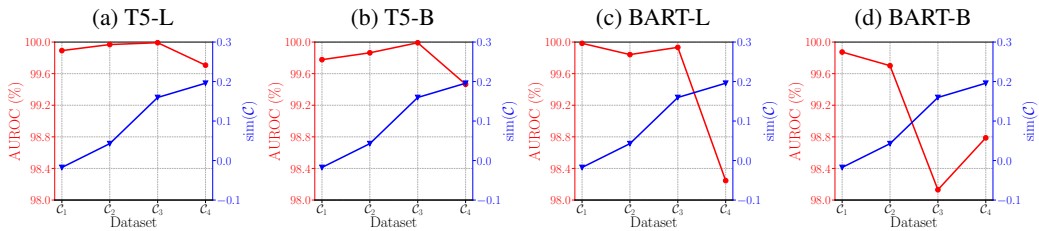

Figure 22: AUROC vs. $\text{sim}(\mathcal{C})$; The coherent paragraphs sourced from the S2orc domain, and varying outlier domains represented as $\mathcal{C}_1$ through $\mathcal{C}_4$, encompassing SAMSum, CNN/Daily Mail, Random Domain, and Delve.

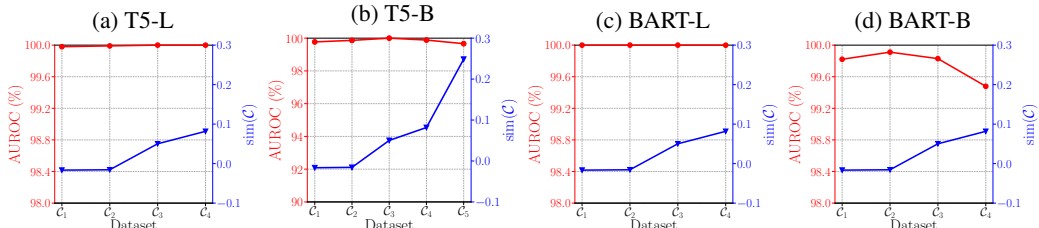

Figure 23: AUROC vs. sim($\mathcal{C}$); The coherent paragraphs sourced from the SAMSum domain, and varying outlier domains represented as $\mathcal{C}_1$ through $\mathcal{C}_4$, encompassing Delve, S2orc, Random Domain, and CNN/Daily Mail.

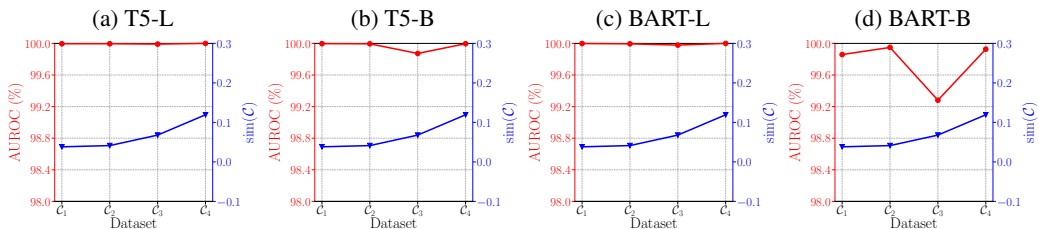

Figure 24: AUROC vs. sim($\mathcal{C}$); The coherent paragraphs sourced from the CNN/Daily Mail domain, and varying outlier domains represented as $\mathcal{C}_1$ through $\mathcal{C}_4$, encompassing Delve, S2orc, SAMSum, and Random Domain.

# I HYPER-PARAMETER SENSITIVITY

In this section, we show how different choice of the hyper-parameter $\alpha$ and $\beta$ affects the cross-document outlier detection performance of our method. Specifically, we present the relationship between the selection of $\alpha$ and $\beta$ and outlier paragraph detection performance. Each figure in this section displays FPR at 95% TPR or AUROC of our method on each dataset and model when selecting different combinations of $\alpha$ and $\beta$. The details of hyper-parameters can be found in Table 11 in D.

We observe that the best performance occurs near $\alpha = 0.6$ for most choices of $\beta$ and the best performance occurs near $\beta = 0.2$ for most choices of $\alpha$. We also observe that the performance does not change much when $\alpha$ varies from 0 to 1. Similarly, the performance also changes slightly when $\beta$ varies from 0 to 0.4. We observed that the performance of CODE on both types of pre-trained models is more sensitive to $\alpha$ compared to $\beta$.

The correspondence between the figures and the setting is as follows:

- Figure 25: FPR at 95% TPR on Delve (1K) dataset and each model.

- Figure 26: FPR at 95% TPR on Delve (8K) dataset and each model.

- Figure 27: FPR at 95% TPR on S2orc dataset and each model.

- Figure 28: FPR at 95% TPR on SAMsum dataset and each model.

- Figure 29: FPR at 95% TPR on CNN/Daily Mail dataset and each model.

- Figure 30: AUROC on Delve (1K) dataset and each model.

- Figure 31: AUROC on Delve (8K) dataset and each model.

- Figure 32: AUROC on S2orc dataset and each model.

- Figure 33: AUROC on SAMsum dataset and each model.

- Figure 34: AUROC on CNN/Daily Mail dataset and each model.

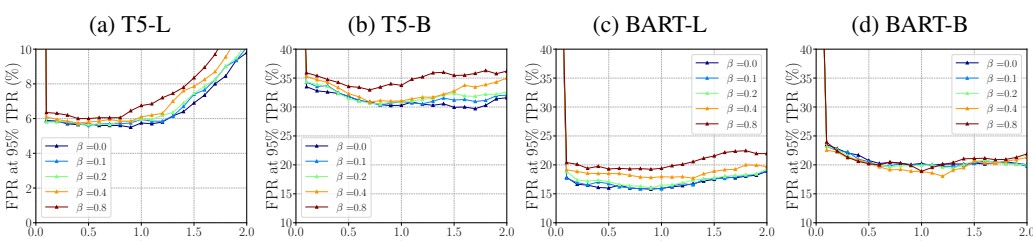

Figure 25: FPR at 95% TPR vs. Hyper-parameter on Delve (1K)

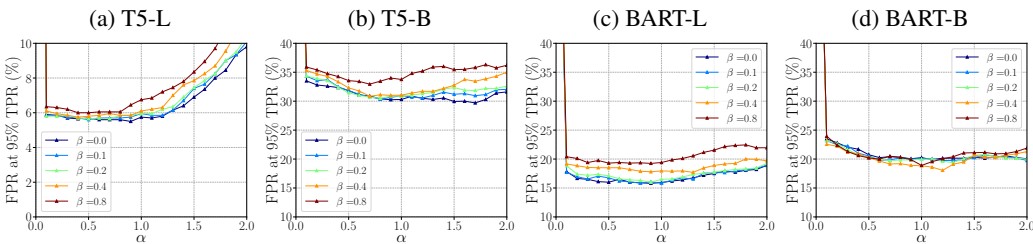

Figure 26: FPR at 95% TPR vs. Hyper-parameter on Delve (8K)

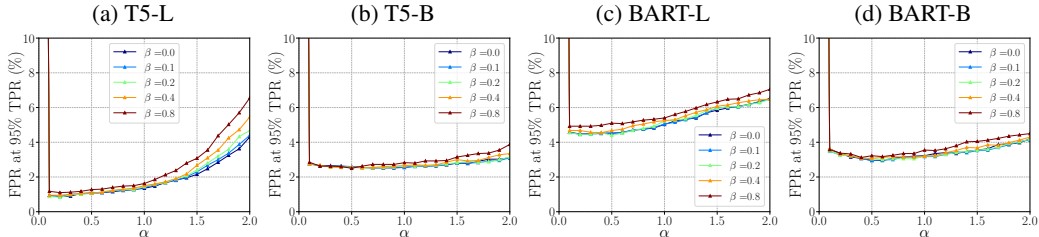

Figure 27: FPR at 95% TPR vs. Hyper-parameter on S2orc

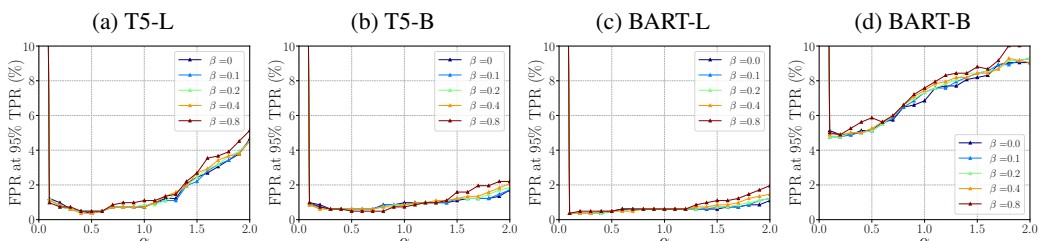

Figure 28: FPR at 95% TPR vs. Hyper-parameter on SAMsum

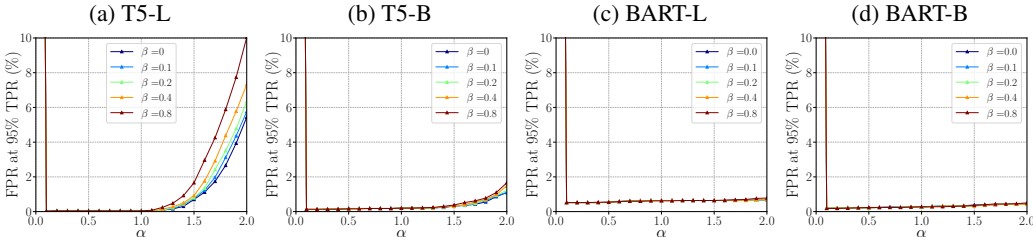

Figure 29: FPR at 95% TPR vs. Hyper-parameter on CNN/Daily Mail

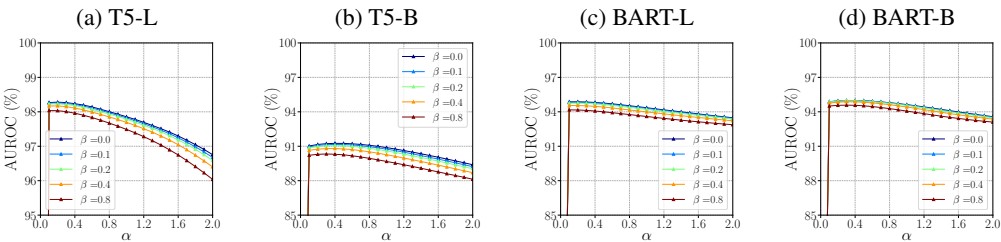

Figure 30: AUROC vs. Hyper-parameter on Delve (1K)

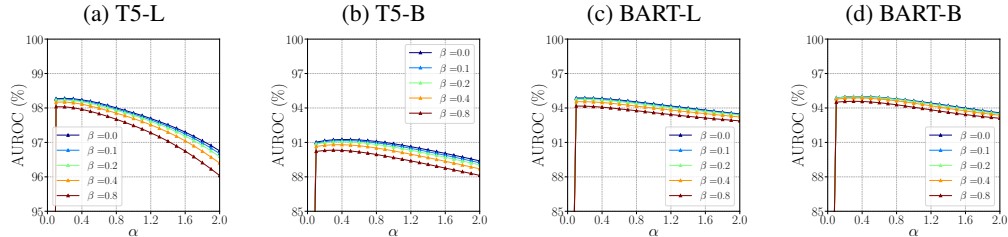

Figure 31: AUROC vs. Hyper-parameter on Delve (8K)

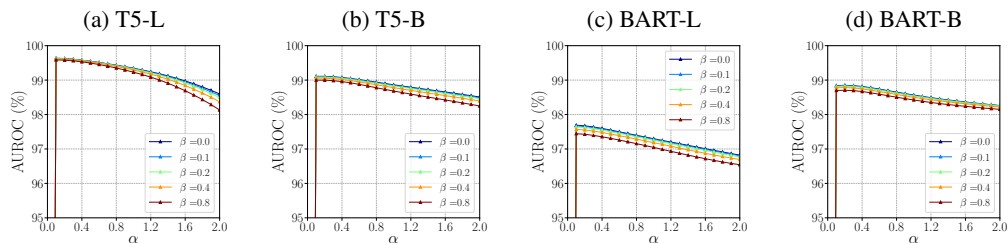

Figure 32: AUROC vs. Hyper-parameter on S2orc

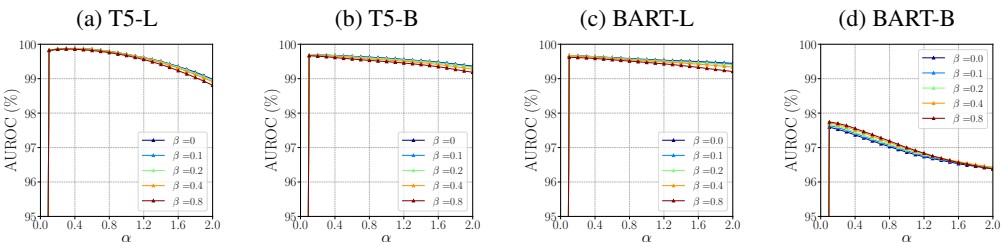

Figure 33: AUROC vs. Hyper-parameter on SAMsum

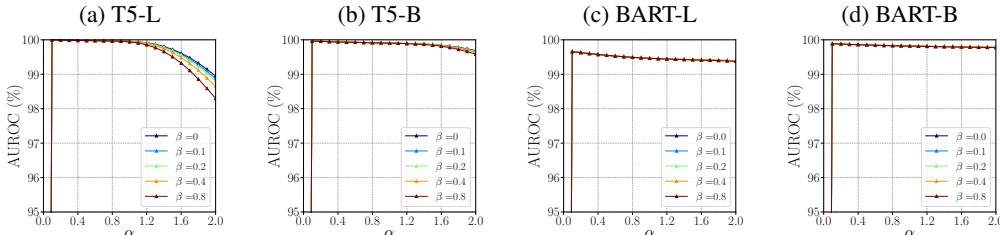

Figure 34: AUROC vs. Hyper-parameter on CNN/Daily Mail

# J SUPPLEMENTARY MATERIAL FOR EFFECTIVENESS OF OUTLIERS IN PRE-TRAINING

In this section, we study how the outlier paragraphs in the pre-training affect the performance. Specifically, we pre-trained the T5-Large model using only coherent paragraphs from the Delve dataset.

We evaluate the pre-trained models with three metrics for text summarization, and Table 12 presents the results. We observe that outlier paragraphs can slightly improve the generation performance. This may be due to the fact that outlier paragraphs may help enrich the corpus in that domain, therefore enhancing the summarization performance.

Table 12: Performance of Pre-trained Model vs. outlier paragraphs

|  |  | **ROUGE-1** | **ROUGE-2** | **ROUGE-L** |
|---|---|---|---|---|
| **outlier paragraphs** | With | 19.34 | 3.38 | 14.42 |
|  | Without | 17.00 | 2.45 | 12.87 |

Table 13 presents three metrics of outlier paragraph detection under the case where T5-Large is pre-trained with and without outliers. We observe that outlier paragraphs plays an important role for outlier detection task.

Table 13: Performance vs. outlier paragraphs (%)

|  |  | **FPR at 95% TPR** | **AUROC** | **AUPR** |
|---|---|---|---|---|
| **outlier paragraphs** | With | 5.80 | 98.08 | 97.03 |
|  | Without | 80.45 | 62.92 | 66.99 |

