# OpenReview forum: "Revealing The Intrinsic Ability of Generative Text Summarizers for Outlier Paragraph Detection"
_ICLR.cc/2024/Conference — Submitted to ICLR 2024_

### Official Review · Reviewer_bZd9 · 2023-10-31

**Soundness:** 2 fair
**Presentation:** 1 poor
**Contribution:** 1 poor
**Rating:** 3
**Confidence:** 3

**Summary:**

This paper investigates the outlier paragraph detection in summarization. It proposes CODE which uses the cross attention between the input paragraph and the generated summary to calculate the relevance.

**Strengths:**

* The proposed CODE is finetune-free and only has two hyperparameters to set.
* It discusses two settings for outlier detection in summarization and conducts a detailed analysis of the influence of each hyperparameter.

**Weaknesses:**

* The problem setting is strange to me. (1) In actual work scenarios, are there many outlier paragraphs in the document? If so, the proposed method should be evaluated on real cases instead of just synthesis data. If outlier paragraphs are uncommon, there would be little point in exploring the issue. It will be better to include more related work on this topic (2) The proposed method and baselines all use the generated summary to identify the outlier. However, If we already have the summary generated, what is the meaning of detecting the outlier paragraphs to avoid its influence on summarization? If the detection of outlier paragraphs will be further used to refine the summary, it should be better clarified in the paper.
* The baselines are too simple and the discussion about related work is missing.

**Questions:**

* The discussion about related work is missing
* The citation of ROUGE is wrong
* some terms are weird
  * neuron output => hidden states / outputs
  * ReLU neurons => ReLU activation function
  * evaluation loss => validation loss
  * Generative Language Model (GLM) is not a commonly used abbreviation and is rarely used to refer to an encoder-decoder based Transformer.

---

### Official Review · Reviewer_tMsa · 2023-11-01

**Soundness:** 3 good
**Presentation:** 2 fair
**Contribution:** 1 poor
**Rating:** 1
**Confidence:** 4

**Summary:**

This paper addresses the problem of paragraph outlier detection in the context of summarization. This problem is about detecting paragraphs inside the original document which are out-of-domain. This is done by computing a similarity score on top of the cross attention of the generated summary and each one of the original paragraphs, and flagging a paragraph as outlier if this below a certain threshold

**Strengths:**

* Extensive experiments
* The proposed approach achieves scores which seem satisfactory

**Weaknesses:**

The biggest weakness is the lack of motivation of this problem. Why is this interesting? Is there any real use-case scenario of this? This is evidenced by the fact that the studied datasets are artificially created to fit exactly this setting.
Coming up with synthetic datasets is a great way of studying problems... if there is at least an attempt made to argument the interest of that problem. This is never done in this paper and the problem is just given as if it were a real one.

Beyond that, there does not seem anything specific to summarization - it seems that this could be applied to any seq2seq problem, in particular translation.

**Questions:**

Please motivate the problem

What is the impact on the summarization quality after this noise is introduced?

---

### Official Review · Reviewer_2Dc8 · 2023-11-01

**Soundness:** 2 fair
**Presentation:** 3 good
**Contribution:** 1 poor
**Rating:** 3
**Confidence:** 4

**Summary:**

This paper investigates the issue of outlier paragraphs, which according to the authors, is detrimental to summarization systems. To address the issue, the authors propose a method to create summarization pre-training dataset that contain random cross-document and cross-domain outlier paragraphs. After training BART and T5 on those synthetic summarization datasets, they propose different methods to detect outlier paragraphs. Specifically, they find that a closed-form detector (CODE), a function of cross-attention and word frequencies, outperforms a model based on the fine-tuning of a MLP classifier head. Extensive analysis of hyperparameters, models, and pre-training settings is presented as evidence of the effectiveness of their method.

**Strengths:**

The paper present a clear formalization of the dataset construction and provides comprehensive empirical investigation of the problem, covering four summarization datasets.

**Weaknesses:**

The paper has important weaknesses related to the relevance of the proposed problem and soundness of the methods. First, the authors claim that summarizers are vulnerable to outlier paragraphs, but they do not cite any previous work supporting this claim and do not provide empirical evidence that the summarization datasets in question actually contain outlier paragraphs (in their original distribution) or measure the effects of outlier content in summarization performance. Instead, the authors create a artificial dataset for which each input sample contains 2 coherent and 2 outlier paragraphs and make their conclusions based on this synthetic setting, focusing on the accuracy of outlier classification.

Importantly, when the language models are trained to summarize those synthetic documents, they are indirectly trained to ignore paragraphs that are uninformative for the summary generation, and thus, it is not surprising that cross-attention weights would be lower for the outlier sentences and that a closed-form solution could capture this pattern. Therefore, the paper does not measure intrinsic properties of existing summarizers (as the title suggests), but only verifies that the models learned to assign more weight to sequences of tokens that are (trivially) more informative for the task.

Typos:
- "GLM-based Outlier _Pargraph_ Detection Problem", page 3
- "inside the GLM with a multi-layer _perception_", page 3
- "Pre-training Summerizers", page 5

**Questions:**

- In page 2, the construction of the dataset is formulated as a paragraph sampling process $P(X | D_i, D'_i)$, which means that there is a probability (function of $\epsilon_i$, which you describe as "small") that the paragraph sequence has no outliers. But in Section 4.2 you state that each data sample contains two coherent paragraphs, two outlier paragraphs, and one summary. Could you clarify this apparent inconsistency?

- Do you have any evidence the issue of outliers is relevant outside your controlled summarization setting? If so, what is the impact of your detector to the task of interest (summarization)?

- It seems that your MLP-based detector loses significant information (by mean pooling embeddings). In contrast, CODE has access to word level cross-attention and frequencies. Do you think they are fairly comparable?

---

### Official Review · Reviewer_G5oa · 2023-11-01

**Soundness:** 2 fair
**Presentation:** 2 fair
**Contribution:** 2 fair
**Rating:** 3
**Confidence:** 3

**Summary:**

The paper proposes CODE, a novel outlier detector using a closed-form expression rooted in cross-attention scores. This can detect both cross-document and cross-domain outliers from a given input set of passages. Results show that CODE is superior under different datasets and architectures.

**Strengths:**

* The paper includes extensive experiments with several ablations that support their hypothesis.

* The use of outlier detection in summarization models may help improve their performance, although this was not explored in the paper.

**Weaknesses:**

* This statement in the introduction needs evidence: "these models struggle with outlier paragraphs interspersed within the content". This could be in the form of an experiment, where the summarization performance (especially in terms of relevance) increases when outlier detection is applied (e.g., by removing the outliers from the input). Without this, it is difficult to find motivations for such a task. Given how cross-document and cross-domain outliers are defined, I don't think these outliers actually exist naturally.

* The datasets are synthetically created, which introduces a concern about whether or not this actually helps in real-world settings. Further experiments on how the outlier detection model affects the current summarization datasets (not the synthetic versions). How many of the input passages actually considered outliers by the model? Are these predicted outliers reasonable predictions?

* What is the rationale behind the use of CNN/DM, SAMSum, Delve, and S2orc in your experiments? Do these datasets have naturally occurring outliers?

**Questions:**

See weaknesses above.

---

### Meta-Review · Area_Chair_m6a7 · 2023-12-12

**Metareview:**

This paper addresses the presence of outlier paragraphs in generating text summaries. To this end, they propose an outlier detector based on cross-attention and word frequencies which they show outperforms LMs finetuned to a synthetic dataset they create, containing cross-document and cross-domain outlier paragraphs.

Strengths: The paper presents extensive experimentation to explore their proposed hypothesis.

Weaknesses: The paper does not convince readers that the presence of outliers is a problem in text summarization. This is exacerbated because the empirical results are presented on a synthetic benchmark. The method uses the summary to detect the outlier paragraphs in the original document, making the logic somewhat circular and bringing to question if the problem is well motivated. There is also very limited discussion of related work.

**Justification For Why Not Higher Score:**

Please see the weaknesses above; lack of clear problem motivation.

**Justification For Why Not Lower Score:**

N/A

---

### Decision · Program_Chairs · 2024-01-16

Reject